# An Intensified Marine Predator Algorithm (MPA) for Designing a Solar-Powered BLDC Motor Used in EV Systems

**Rajesh Kanna Govindhan Radhakrishnan** [1], **Uthayakumar Marimuthu** [2,3], **Praveen Kumar Balachandran** [4], **Abdul Majid Mohd Shukry** [3] **and Tomonobu Senjyu** [5,*]

1    Department of EEE, Kalasalingam Academy of Research & Education,
     Virudhunagar 626125, Tamil Nadu, India
2    Department of Automobile Engineering, Kalasalingam Academy of Research & Education,
     Virudhunagar 626125, Tamil Nadu, India
3    Faculty of Mechanical Engineering & Technology, University Malaysia Perlis (UniMAP),
     Arau 02600, Perlis, Malaysia
4    Department of EEE, Vardhaman College of Engineering, Hyderabad 501218, Telangana, India
5    Faculty of Engineering, University of the Ryukyus, Okinawa 903-0213, Japan
*    Correspondence: b985542@tec.u-ryukyu.ac.jp

**Abstract:** Recently, due to rapid growth in electric vehicle motors, used and power electronics have received a lot of concerns. 3φ induction motors and DC motors are two of the best and most researched electric vehicle (EV) motors. Developing countries have refined their solution with brushless DC (BLDC) motors for EVs. It is challenging to regulate the 3φ BLDC motor's steady state, rising time, settling time, transient, overshoot, and other factors. The system may become unsteady, and the lifetime of the components may be shortened due to a break in control. The marine predator algorithm (MPA) is employed to propose an e-vehicle powered by the maximum power point tracking (MPPT) technique for photovoltaic (PV). The shortcomings of conventional MPPT techniques are addressed by the suggested approach of employing the MPA approach. As an outcome, the modeling would take less iteration to attain the initial stage, boosting the suggested system's total performance. The PID (proportional integral derivative) is used to govern the speed of BLDC motors. The MPPT approach based on the MPA algorithm surpasses the variation in performance. In this research, the modeling of unique MPPT used in PV-based BLDC motor-driven electric vehicles is discussed. Various aspects, which are uneven sunlight, shade, and climate circumstances, play a part in the low performance in practical scenarios, highlighting the nonlinear properties of PV. The MPPT technique discussed in this paper can be used to increase total productivity and reduce the operating costs for e-vehicles based on the PV framework.

**Keywords:** brushless DC motor (BLDC); maximum peak point tracking (MPPT); photovoltaic (PV) systems; electric vehicle (EV) applications; marine predator algorithm (MPA)

## 1. Introduction

The advancement of electric vehicles is driven by the ambition to reduce emissions to increase the consumption of fuels [1]. In this situation of India and China, the shortage of energy is anticipated to happen rapidly because of a reduction in the future availability of fossil fuels and a 76% hike in necessity in the period from 2020 to 2045 [2]. Carbon emissions and waste are decreased by employing renewable energy [3]. Due to this, there is an increase in demand for clean, pollution-free renewable energy that emits only 30 carbon dioxides [4]. Industry and researchers have utilized advanced PV modules for many purposes due to consecutive reductions in the price of PV panels and power electronics components [5]. To maximize a PV array's capacity, the MPPT approach with a DC-DC converter topology is commonly utilized [6]. No carbon emissions are produced. Industry and researchers have utilized the advanced solar PV array for many applications because

of continuous reductions of price in power electronics components and PV panels [7]. The MPPT approach with DC-DC converters typically maximizes a PV array's capacity [8]. Various MPPT control techniques have been proposed, with fractional open/short-circuit control methods, incremental conductivity (INC), and perturbation and observation (P&O) being the most often used conventional techniques. These techniques result in a high turnout in a steady-state activity [9]. These algorithms were verified to be not effective as when the weather is bad, conversion ratios are slow, and bigger variances prevent them from obtaining an overall maximum power point (MPP) in settings with partial shading conditions. To deal with these problems, MPPT with a bioinspired optimization algorithm has been proposed.

The artificial immune system (AIS) and the metaheuristic genetic algorithm (GA) were applied to overcome such nonlinear uncertain conditions because of the appropriate particular sensor and the complicated circuitry [10]. However, immune cells have a huge population structure and adaptive machinery, which results in a poor conversion rate and a lengthy conversion time for AIS and GA algorithms [11]. Crossover procedures are used in conjunction with computational convergence time to enhance the mutation. Many MPPT techniques with bioinspired optimization are implemented to challenge such difficulty [12]. FSA is a fish life-inspired methodology designed to reduce grade point average assessment (GMPP) oscillations. Numerous control settings are needed for PSO's random accelerating value choosing, and it may be a significant drawback. The bioinspired optimization techniques presently have more tracking efficiency, a high convergence rate, and low transients [13]. Gray wolf (GW), ant colony (AC), glowworm optimization algorithm, and fish swarm algorithm (FSA) are a few examples. However, due to less bee availability and the weather being unpredictable, the poor conversion rate in ABC approaches [14,15].

Due to a shortfall of contingency and a heavy nest population, the cuckoo search algorithm is a more productive way for nonlinear-based issues, although its rate of melting is moderate. Due to this, several researchers have implemented this bioinspired approach based on photovoltaic system investigation. Considering the difficulty present in MPPT techniques, this paper proposed a novel MPPT control technique for MPA. It does not need hardware data from PV, as it can exactly and rapidly search to find the GMPP. This work object is to enhance the overall performance of PV-powered electric vehicles. In this technology, the BLDC motor is used in PV-powered e-vehicles. The MPA technique has been implemented to increase the complete performance of the system. The MPA technique features a faster convergence rate and a better method to locate GMPP. An observation of MPPT output has been illustrated to determine the effectiveness of the suggested approach in this framework.

For maximum power tracking from solar PV, making use of combined MPPT, different techniques were presented by scientists. In the following segment, the different MPPT approach for EVs driven by BLDC is surveyed, and it was designed for maximum power tracking.

The complete paper's structure is provided below: The literature review is presented in Section 2. the proposed work is explained in Section 3. Sections 4 and 5 give the control techniques which were used in the proposed work, followed by its results in Section 6 and conclusion in Section 7.

## 2. Literature Review

Himabindu et al. [16] presented the partially solar-powered EV. The EV's energy efficiency is greater than that of fuel-powered vehicles without taking electricity generation, transmission, and efficiency into account. Moreover, for a limited solar-powered EV, the unique prototype of a lightweight EV was elaborated on in this paper. The development of the unique energy-efficient prototype of EV and the possibility of a limited solar-based EV was discussed. Lakshmiprabha et al. [17] presented the BLDC motor with a PV-based electric vehicle approach. The approach for developing the BLDC driven with PV-powered EV, which was a potential solution for the lake of impending, was explained in this work.

The approaches to finding the right parts of this application were explored, and both of them were tested and simulated in a real-world application. The integrated system of the PV-powered EV features the BLDC motor, batteries, battery charger controller, solar module, and a DC-DC boost converter. Ahmad et al. [18] demonstrated that the nature of the autoindustry was changing as a result of worries about oil supply, foreign relations, and fuel prices. There were numerous hybrid technologies available at the time, due to the availability of hydrogen. Among the oldest vehicles using alternative fuel, the vehicle integrated with solar power has several applications in the expanding EV market. The development of the solar-powered telemetry system for high-speed cars helps in improving the understanding of the vehicle's power aspects and the operation implemented in EVs. This work inspected the position and history of electric vehicles and solar energy, in addition to a standard solar vehicle.

García et al. [19] conferred on e-rickshaws driven by a BLDC motor a fuzzy logic controller (FLC)-based technique to develop ideal power management for regenerative braking. The FLC was adapted to control the separate power management for the battery and for the supercapacitor, to supply the output of the e-rickshaw driven by BLDC. E-rickshaw enhanced operating time by the solar-powered approach to boost the operation, and using simulated testing rickshaws was verified, which exposed the examination of the BLDC's performance under several operating conditions. If the need for power increases suddenly in a temporary situation, the supercapacitor manages the complete need for power. The power ratio is divided to enable the battery to be deeply discharged, increasing battery life. Ho et al. [20] explained the integration of electric power systems for the EV. The objective of this work was to introduce the theoretical arrangement to successfully integrate EVs into electric interconnected networks. The advanced structure was split into power market environments and the grid technical operations. Participants in both processes, as well as their actions, were all considered and fully explained. Moreover, various simulations, with the dynamic and analysis of steady-state behavior, were explained to make clear the impacts and benefits originating from the EVs and integration of the grid using the cited methodology. Oubelaid et al. [21] demonstrated the controlling techniques for hybrid electric vehicles. Global optimization techniques and dynamic programming were mainly employed to evaluate the powertrain configuration's prospective fuel efficiency. These control procedures cannot be applied directly until advanced driving conditions could be likely at the time of real-world application; even so, the results obtained with this noncausal method delivered the criteria for analyzing the best possible control technique that is attainable.

Lan et al. [22] conferred the creation of the Japanese government's EV policy. The scope of this work was to inspect the policy for the creation of alternative vehicles to traditional vehicles, the outcomes of government actions, and the requirement of a technological adaptability program supported by the government. The effects of this scheme on the methods of innovation were explored through the use of this viewpoint and technological literature improvement. The complete network with the assistance was investigated, further to the context in which this different policy has been used since the early 1970s. Saha et al. [23] demonstrated for EVs with BLDC Motors that are electric, hybrid, and plug-in hybrid an effective regenerative braking system using battery/ultracapacitor. The ultracapacitor used a suitable inverter switching template for energy regeneration and/or regenerative braking to store the vehicle's kinetic energy. Due to this, no extra power electronics interfaces were needed. Simultaneously, the EV's front and back wheels received braking force from the artificial neural network controller, which is responsible for distributing it. To attain steady torque braking, additionally, the PI controller was used to vary the PWM operating cycle. Li et al. [24] explained that the BLDC motors are controlled by a hybrid sliding-mode system without a position sensor (HSMC). This research gave effective and reliable control techniques for the position-sensorless EV using the BLDC motor. To adopt the BLDC motor sensorless control of the BLDC motor, the back EMF finding technique was initially implemented and enhanced. The corresponding circuits of regulating systems were presented,

as well as the creation of energy regeneration and standard driving mathematical models. A technique for the EV HSMC approach was implemented to promise by integrating both the system effectiveness and using the high-order sliding-mode approach; the nonsingle terminal sliding mode has sustained stability.

Gupte et al. [25] have conferred on transmission a selectively aligned surface (PM-BLDC) for the HEV motor. A programmable and adjustable generated voltage constant in an axial-flux PMBLDCM was used to achieve the field weakening. This quality was exclusively suitable in motors for driving vehicles with vast ratios of constant-power speeds, where it was imperative to get rid of gear shifts and shrink the overall motor drive's size. The advantages of this method's high pole count were discussed, and the simulation's impact on the kilovolt-ampere motor drive, acceleration, maximum speed, and efficiency was described over regular driving cycles. The e-vehicles with BLDC motors used in this system are energized using solar PV. The MPA technique is derived concerning enhancing the system's total efficiency. For efficient MPP tracking from a PV array, scientists took help of the MPA technique.

### 3. Modeling and Description of the Entire System

Figure 1 illustrates the BLDC motor with a PV-powered, battery-operated architectural arrangement for EV applications. From left to right in Figure 1, the system consists of a solar PV array, a DC-DC converter, a battery, a VSI, and a BLDC motor. The PV array produces power and is given as an input to the DC-DC boost converter, which utilizes the MPA technique to operate the MOSFET switch. To get more power output from the PV array, the MPPT algorithm is implemented. The boost converter is used to provide the essential power for battery charging; the PV is serially connected to the battery bank, to operate the BLDC motor through an inverter. The electronics commutation is used to generate the switching pulse required for an inverter connected with a BLDC motor. In the following sections, we will discuss how to create and control the suggested system.

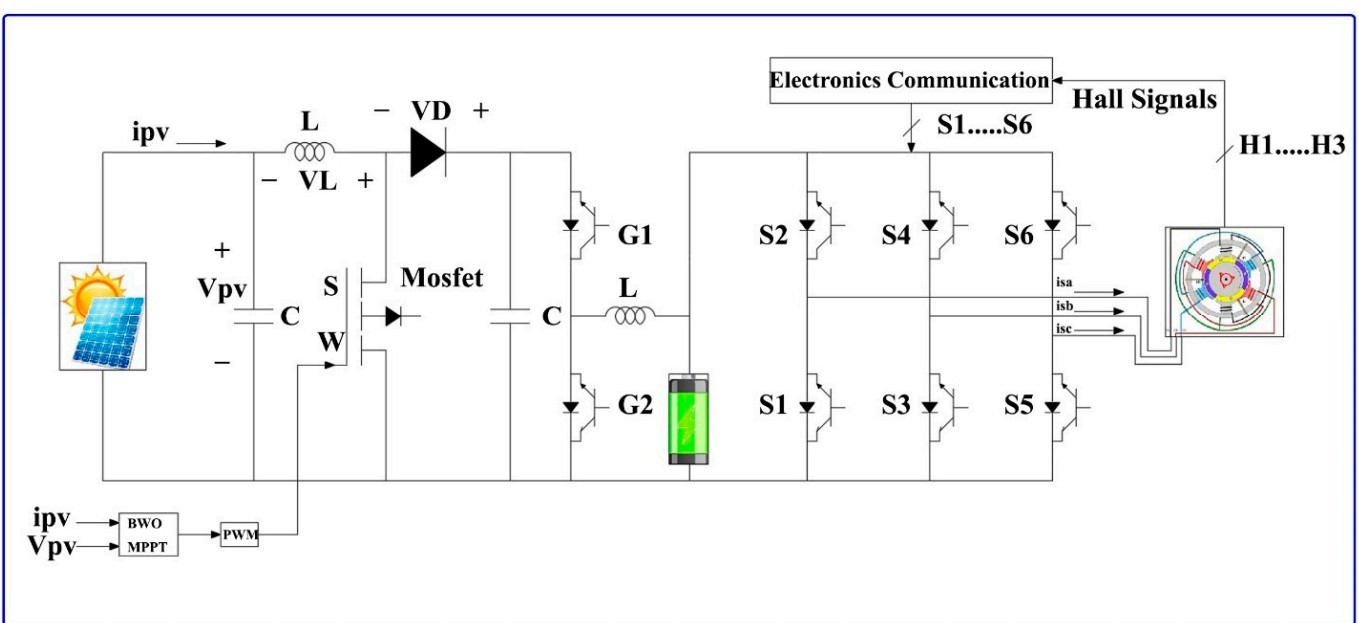

**Figure 1.** Circuit arrangement for the advanced battery-operated, solar PV-powered, BLDC motor-operated system.

### 3.1. Proposed System's Design Configuration

In Figure 1, the alignment of the suggested system configuration has been presented. The most essential parts of the system configuration are the SPV array, DC-DC converter,

battery bank, and brushless DC. This system is built to function satisfactorily regardless of changes in the amount of solar irradiation.

### 3.2. Arrangement of Solar PV Array

The number of PV modules connected in parallel and series in a solar array is used to estimate the current, voltage, open-circuit voltage, and short-circuit current. The comparable circuit of PV cells is illustrated in Figure 2. The parallel diode, current source, and series resistors are the components required. To create photovoltaic modules, the PV cells are built simultaneously. The required power is based on a combination of parallel and series supply. U* a and U* p represent the number of parallel and series photovoltaic cells, respectively. The voltage and current output relationship can be expressed as

$$I'_{PV} = N'_P I'_G - N'_P I'_S \left( \exp\left[ \frac{q^*}{AKT_C} \left( \frac{V'_{PV}}{N'_S} + \frac{R_S I'_{PV}}{N'_P} \right) \right] - 1 \right) \tag{1}$$

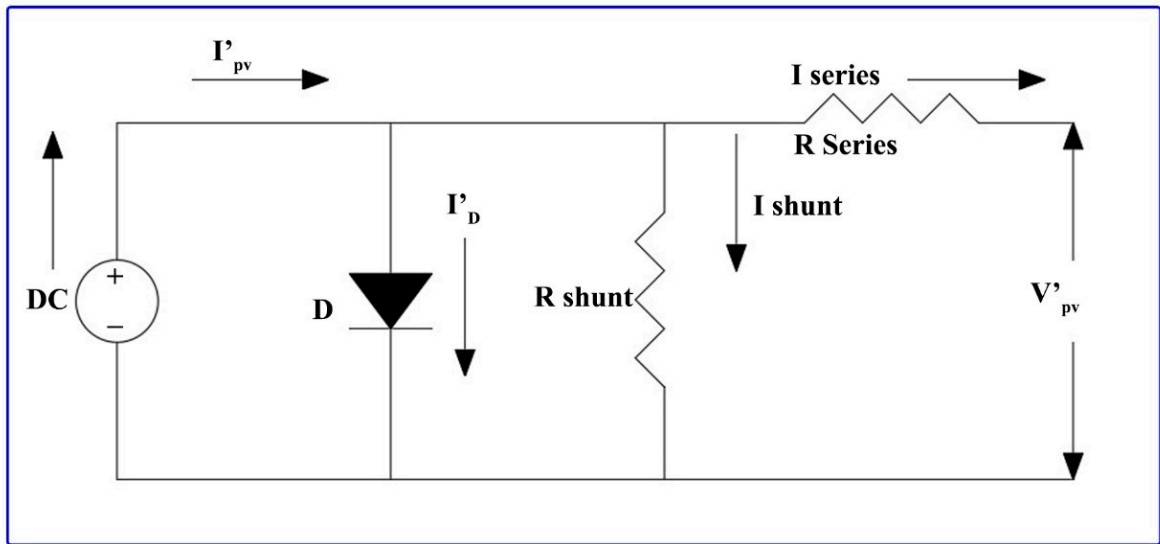

**Figure 2.** PV cell equivalent circuit.

Photocurrent $I'_G$ is produced by solar irradiation, as shown below:

$$I'_G = I'_{sc} + k_1 \left( T_C - T_{ref} \right) \frac{S}{1000} \tag{2}$$

$I'_S$ is explicit as the PV cell saturation current and temperature variation based on the following relationship:

$$I'_S = I'_{rs} \left[ \frac{T_C}{T_{ref}} \right]^3 \exp\left[ \frac{q' E_G}{AK} \left( \frac{1}{T_{ref}} - \frac{1}{T_C} \right) \right] \tag{3}$$

### 3.3. PV Characteristics

The PV array's nonlinear characteristics are dependent on temperature and irradiance. Variation in temperature and irradiation causes a change in them. The V-I and P-V characteristics at different irradiation and constant temperature ($1000 \text{ W/m}^2$, $800 \text{ W/m}^2$, $500 \text{ W/m}^2$, $350 \text{ W/m}^2$) are depicted in Figure 3, as well as the V-I and P-V characteristics at changing temperature and constant irradiation are depicted in Figure 4.

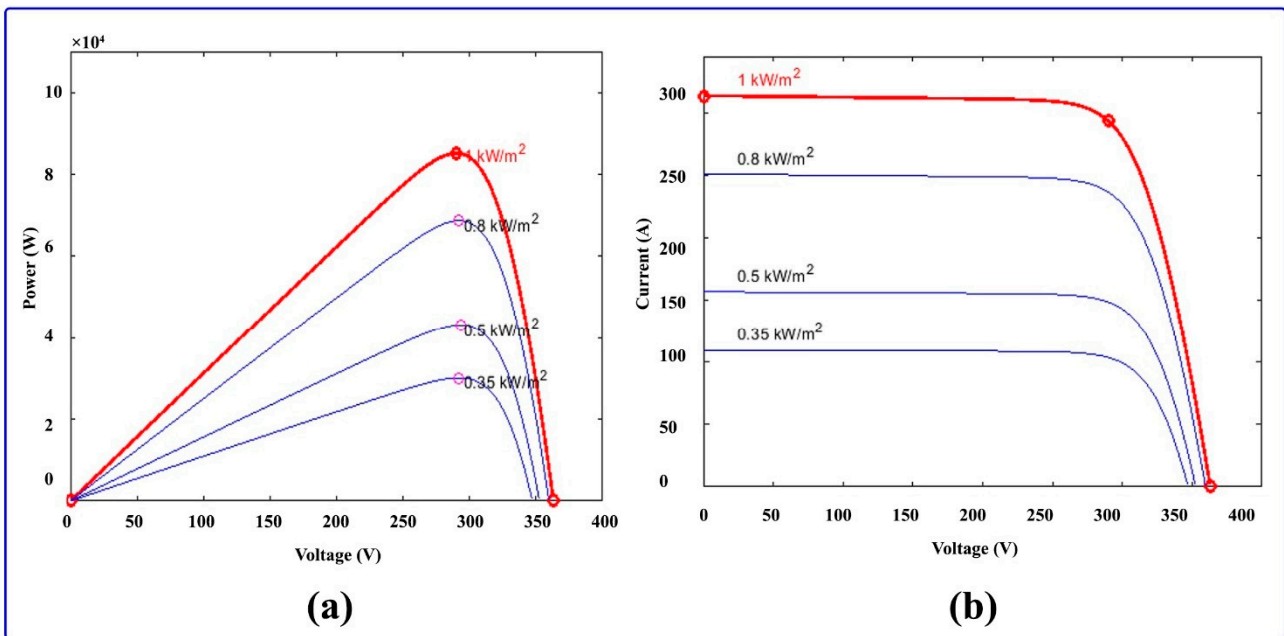

**Figure 3.** (**a**) P-V curve, during constant temperature (**b**), V-I curve, during constant temperature.

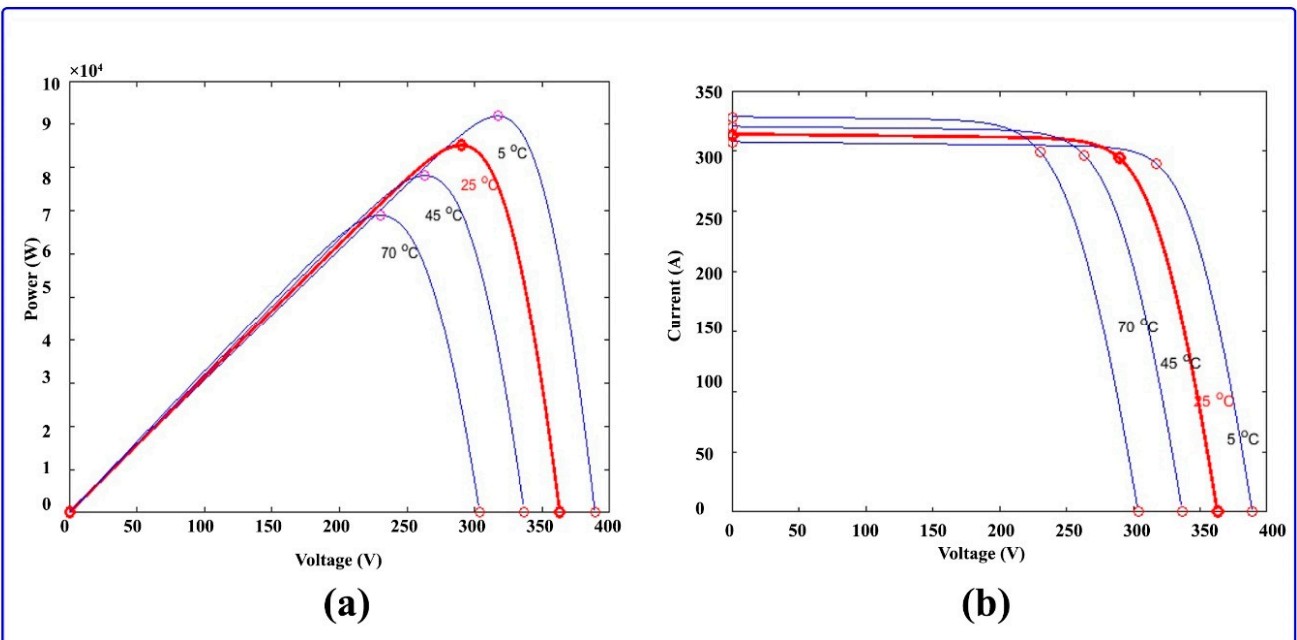

**Figure 4.** (**a**) Continuous irradiation of the P-V curve, (**b**) continuous irradiation of the V-I curve.

*3.4. DC-DC Boost Converter Equivalent Circuit*

The equivalent circuit of the DC-DC converter is shown in Figure 5. At the initial stage Switch, sw1 and sw2 are in closed and open positions, respectively, and the inductor current (IL) will be raised from zero. Consequently, switch sw1 and sw2 are in open and closed positions respectively; at that time, the inductor current will supply the load, and the charges will be stored in the capacitor. The voltage in proportion to the duty cycle of the input and output of a DC-DC converter is depicted in this equation.

$$\frac{V'_O}{V'_{IN}} = \frac{1}{1 - d'_{DUTY}} \tag{4}$$

$$\frac{V'_O}{V'_{IN}} = \frac{T'_{RISE}}{T'_{FALL}} + 1 \tag{5}$$

$$d'_{DUTY} = \frac{T'_{RISE}}{T'_{RISE} + T'_{FALL}} \tag{6}$$

where, $d'_{DUTY}$ = duty cycle, $T'_{RISE}$ = switch sw1 is in closed at the moment of raising the inductor current, $T'_{FALL}$ = switch sw1 is open at the moment when the inductor current is falling.

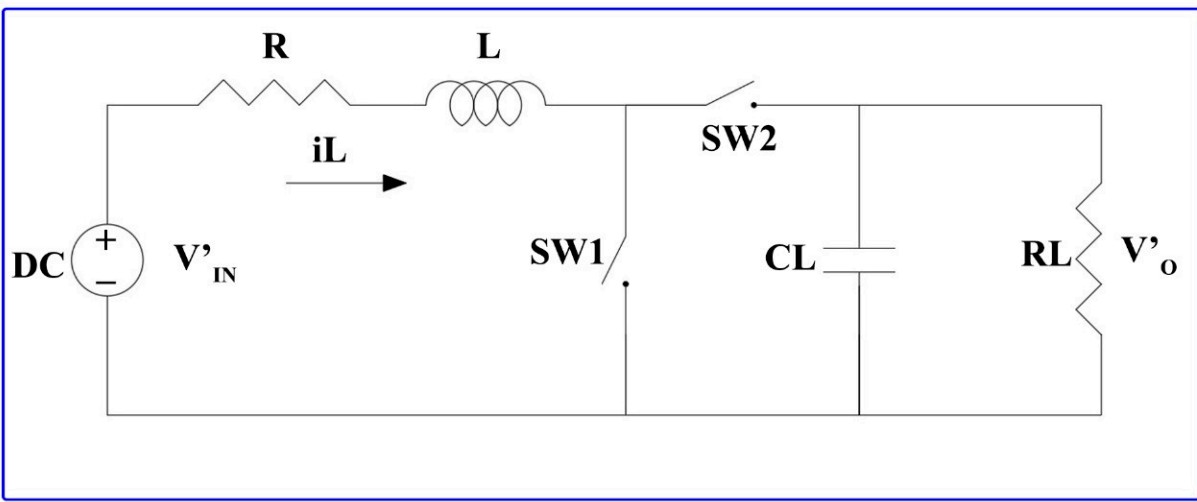

**Figure 5.** DC-DC boost converter equivalent circuit.

### 3.5. BESS Equivalent Circuit

The presented battery equivalent circuit contains polarization capacitor Cpl, polarization resistor Rpl, and ohmic internal resistor R, where the battery transient feedback is simulated using Rpl and Cpl in both charging and draining modes. In that, V (h(t)) represents the nonlinear function of the SoC for h(t)'s. The terminal voltage is taken as the calculated output, and the current is considered as a control input. Figure 6 depicts a lithium-ion battery, and MATLAB/Simulink software is used to perform the simulation. Figure 7 shows the input characteristics of the battery energy storage system where E0 is constant voltage (V), K is polarization constant in (Ah$^{-1}$), A is exponential voltage (V), and B is the exponential capacity (Ah$^{-1}$).

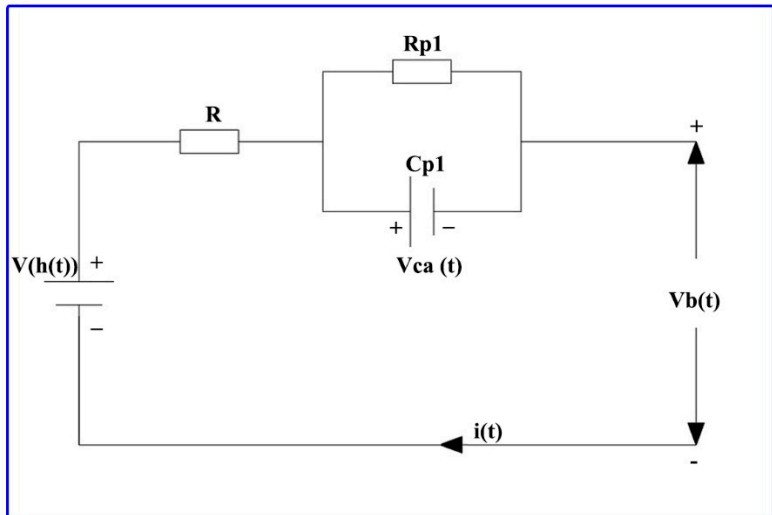

**Figure 6.** BESS equivalent circuit.

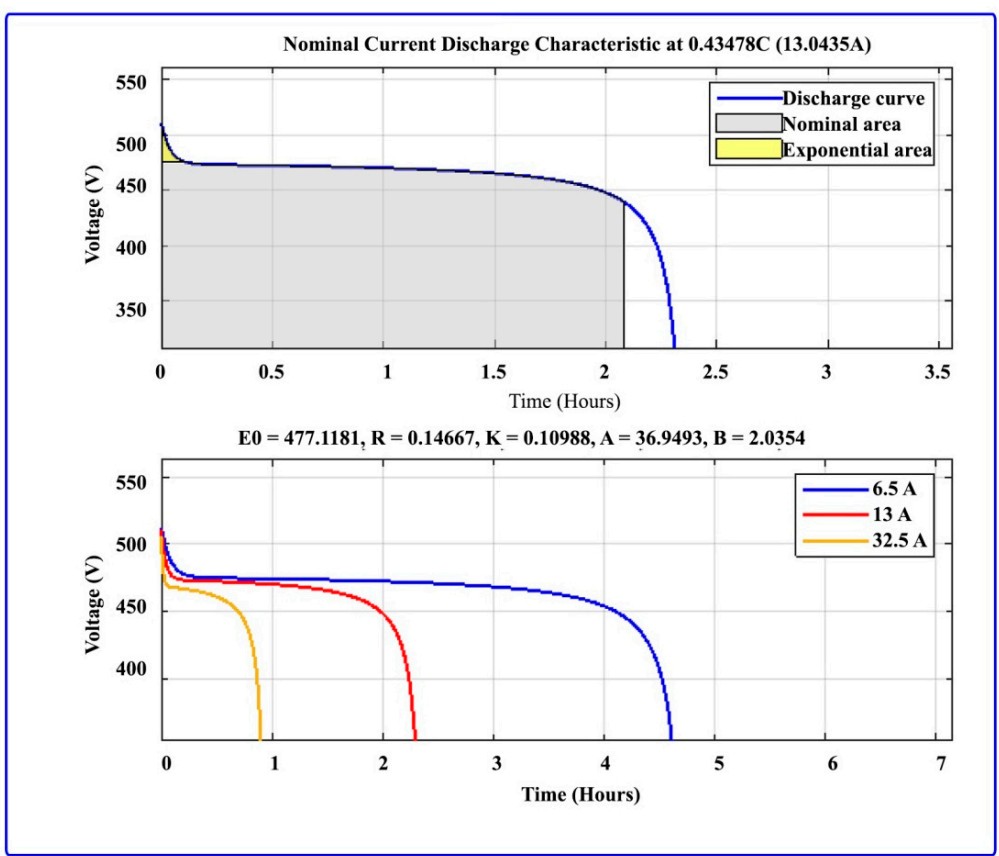

**Figure 7.** Input characteristics of battery energy storage system.

To build the 440 V/30 Ah and a 100% SoC, interconnected modules are used to actualize the battery. The circuit dynamics is expressed by applying Kirchhoff's law:

$$V_b(t) = V(h(t)) - R * i(t) - V_{ca}(t) \tag{7}$$

$$\frac{dV_{ca\,(t)}}{dt} = -\frac{1}{c_{pi}R_{pi}}V_{ca}(t) + i(t) \tag{8}$$

Here, $V_b(t)$ = terminal voltage, $i(t)$ = terminal current, and $V_{ca}(t)$ = voltage across RC, which cannot be directly computed.

### 3.6. Modeling and Motor Choice

In recent years, many electric motors have been used in electric vehicles. In the e-rickshaw, the DC motor's dynamic properties are better; the main disadvantage of the DC motor is that it needs more maintenance due to the brush and commutator. Induction motors are therefore a better option, because they are often suitable for such circumstances, but the induction motor needs huge control. Thus, in automotive applications, the induction motor is not usually employed. After that, the researchers take an alternative and find a trustworthy and effective motor. The BLDC motors are easy to regulate, require less maintenance, and have a high roughness. It has high torque, fast dynamic responses, a low operating voltage range, and a good performance ratio.

The BLDC motor consists of a permanent magnet stator and three-phase windings in the rotor. The currents generated in the rotor can be neglected, and there is no need to model damper windings if the stainless-steel retaining sleeves and magnet have high resistance. The analogous circuit of a BLDC motor is depicted in Figure 8, where R is a stator resistance, L is self-inductance and mutual inductance, and e is phase back-EMF

voltage of A, B, and C, respectively. The 3 φ winding governing equation for the phase variables is

$$\begin{bmatrix} V_a^* \\ V_b^* \\ V_c^* \end{bmatrix} = R^* \begin{bmatrix} 1 & 0 & 0 \\ 0 & 1 & 0 \\ 0 & 0 & 1 \end{bmatrix} \begin{bmatrix} i_a^* \\ i_b^* \\ i_c^* \end{bmatrix} + \begin{bmatrix} l - m & 0 & 0 \\ 0 & l - m & 0 \\ 0 & 0 & l - m \end{bmatrix} \frac{d}{dt} \begin{bmatrix} i_a^* \\ i_b^* \\ i_c^* \end{bmatrix} + \begin{bmatrix} E_a^* \\ E_b^* \\ E_c^* \end{bmatrix} \qquad (9)$$

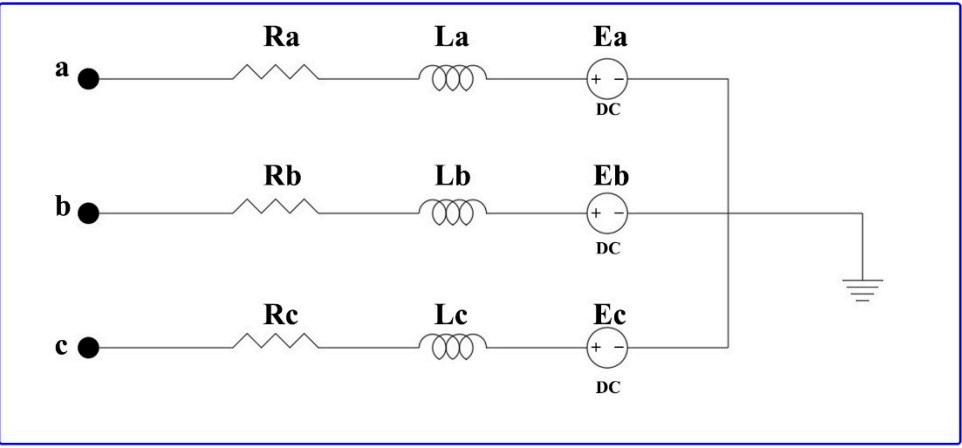

**Figure 8.** BLDC motor equivalent circuit.

$R^*$ = phase resistance, $m$ = mutual inductance, $l$ = phase inductance.
The mechanical equation is shown below:

$$J^* \cdot \frac{d\omega_r^*}{dt} = T_e^* - T_l^* - f_r^* \omega_r^* \qquad (10)$$

Finite element analysis is used to calculate the three back-EMFs, and Fourier series equations are used to display the results. It is a ratio of speeds.

## 4. Control Method Using MPA Technique

As shown in Figure 1, to carry out the required operation and to get the output from PV, MPPT with a DC-DC converter is needed. In the implementation of MPPT, a control variable (duty cycle) is controlled by the MPPT controller. This generates a control signal in the range [0, 1] which is given in Equations (11) and (12):

$$V_{out} = \frac{V_{in}}{1 - d} \qquad (11)$$

$$d = \frac{T_{on}}{T_{Switching}} \qquad (12)$$

where, $V_{out}$ and $V_{in}$ are boost converter output and input voltages, and d denotes the duty cycle. This article gives a new bioinspired algorithm based on marine predators' social behavior pattern.

### 4.1. Marine Predator Algorithm

The marine predator algorithm (MPA) is a bioinspired, metaheuristic optimization technique [26] that has been applied to various optimization problems. A few of the applications of MPA are estimating the parameters of solar PV cells [27], MPPT for solar PV systems [28], and many more. In this section, the MPA is applied in MPPT in an optimized way to the optimal expected output for EVs.

The key points of MPA are (i) the Levy motion for a prey environment of low concentration given in Equation (13), (ii) the Brownian motion for a prey environment of high

concentration given by Equation (14), and (iii) the very decent memory in recalling their partners and the location of successful hunting shown in Figure 9. These features make the marine predator's technique more advanced compared to other bioinspired techniques. The population can be started by Equation (15). Dmin and Dmax are the lower and upper limits for the variables, and rand is the random number.

$$\text{Lévy} \ (\alpha) = 0.05 \times \frac{x}{|y|^{\frac{1}{\alpha}}} \tag{13}$$

$$f(x; \ \mu, \ \sigma) = \frac{1}{\sqrt{2\pi}} e^{-\frac{x^2}{2}} \tag{14}$$

$$D_0 = D_{min} + \text{rand} \ (D_{max} - D_{min}) \tag{15}$$

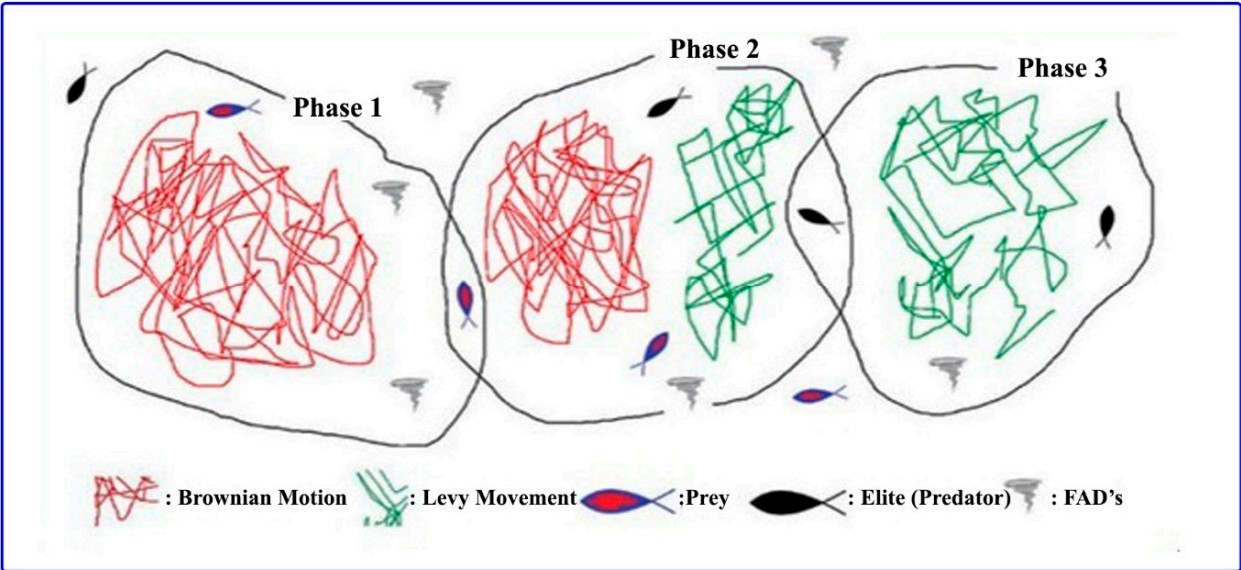

**Figure 9.** Three phases in marine predator algorithm (MPA) optimization.

An elite matrix is developed by the fittest solutions among the marine predators following the survival of the fittest idea. Naturally, the topmost predators (denoted by de) are brilliant in hunting and is given in Equation (16). The position of the predator gets updated from time to time. The prey matrix is developed in which $d_{i,j}$ gives the jth position of the prey and is given by Equation (17).

$$\text{Elite} = \begin{bmatrix} de_{1,1} & \cdots & de_{1,n} \\ \vdots & \ddots & \vdots \\ de_{m,1} & \cdots & de_{m,n} \end{bmatrix}_{m \times n} \tag{16}$$

$$\text{Prey} = \begin{bmatrix} d_{1,1} & \cdots & d_{1,n} \\ \vdots & \ddots & \vdots \\ d_{m,1} & \cdots & d_{m,n} \end{bmatrix}_{m \times n} \tag{17}$$

Optimization Process of MPA

There are three phases in optimization, as shown in Figure 9. Depending upon the velocity ratio and time, the phases are classified. Phase 1: predator is traveling slower than the prey (increased velocity ratio). Phase 2: predator and prey are at the almost same pace (unity velocity ratio). Phase 3: predator is traveling faster than the prey (decreased velocity ratio).

The prey is traveling faster than the predator. This is called as exploration phase, and it happens only in the initial or starting iterations of the algorithm and is given by Equations (18) and (19). Here, R is a rand [0, 1]. Set in a high exploration phase, this phase happens for the first three of the iterations. Prey is accountable for the exploration, and it is given by the Equations (20)–(26). That CF is a step-size controlling parameter for a predator:

$$\overrightarrow{Stepsize_a} = \overrightarrow{R_B} \times \left( \overrightarrow{Elite_a} - \overrightarrow{R_B} \times \overrightarrow{Prey_a} \right); \ a = i \dots n \tag{18}$$

$$\overrightarrow{Prey_a} = \overrightarrow{Prey_a} + P \overrightarrow{R} \times \overrightarrow{Stepsize_a} \tag{19}$$

For the predator population:

$$\overrightarrow{Stepsize_a} = \overrightarrow{R_L} \times \left( \overrightarrow{Elite_a} - \overrightarrow{R_L} \times \overrightarrow{Prey_a} \right); a = i \dots \frac{n}{2} \tag{20}$$

$$\overrightarrow{Prey_a} = \overrightarrow{Prey_a} + P \overrightarrow{R} \times \overrightarrow{Stepsize_a} \tag{21}$$

For the prey population:

$$\overrightarrow{Stepsize_a} = \overrightarrow{R_B} \times \left( \overrightarrow{R_B} \times \overrightarrow{Elite_a} - \overrightarrow{Prey_a} \right); a = \frac{n}{2} \dots n \tag{22}$$

$$\overrightarrow{Prey_a} = \overrightarrow{Elite_a} + P \overrightarrow{CF} \times \overrightarrow{Stepsize_a} \tag{23}$$

$$\overrightarrow{Stepsize_a} = \overrightarrow{R_L} \times \left( \overrightarrow{R_L} \times \overrightarrow{Elite_a} - \overrightarrow{Prey_a} \right); \ a = 1 \dots n \tag{24}$$

$$\overrightarrow{Prey_a} = \overrightarrow{Elite_a} + P \overrightarrow{CF} \times \overrightarrow{Stepsize_a} \tag{25}$$

To avoid the eddy formation or fish aggregating devices (FADs), which may change the marine predators' behavior, marine predators may take a long jump, as is given in Equation (26):

$$\overrightarrow{Prey_a} = \begin{cases} \overrightarrow{Prey_a} + CF \left( \overrightarrow{D_{min}} + \overrightarrow{R} \times \left( \overrightarrow{D_{max}} - \overrightarrow{D_{min}} \right) \times \overrightarrow{U} \right) & \text{if } r \leq FAD_1 \\ \overrightarrow{Prey_a} + (FAD_s \times (1-r) + r) \left( \overrightarrow{Prey_{r1}} - \overrightarrow{Prey_{r2}} \right) & \text{if } r \leq FAD_s \end{cases} \tag{26}$$

### 4.2. Implementation of MPA for MPPT during PSCs

In the search space between Dmin and Dmax [0 to 1], the particles should be initialized for the implementation of MPPT using the MPA optimization technique with a population size of 4, since the work uses four panels in the array. Due to irradiance change, the power will change at that time the code will automatically restart or reinitialize and is given by the conditional Equation (27).

$$\text{if } \frac{\left| P_{PV_{new}} - P_{PV_{old}} \right|}{P_{PV_{old}}} \geq P_{PV}(\%) \tag{27}$$

The flow chart of the proposed bioinspired MPA technique-based MPPT is given in Figure 10, which will validate Equation (27).

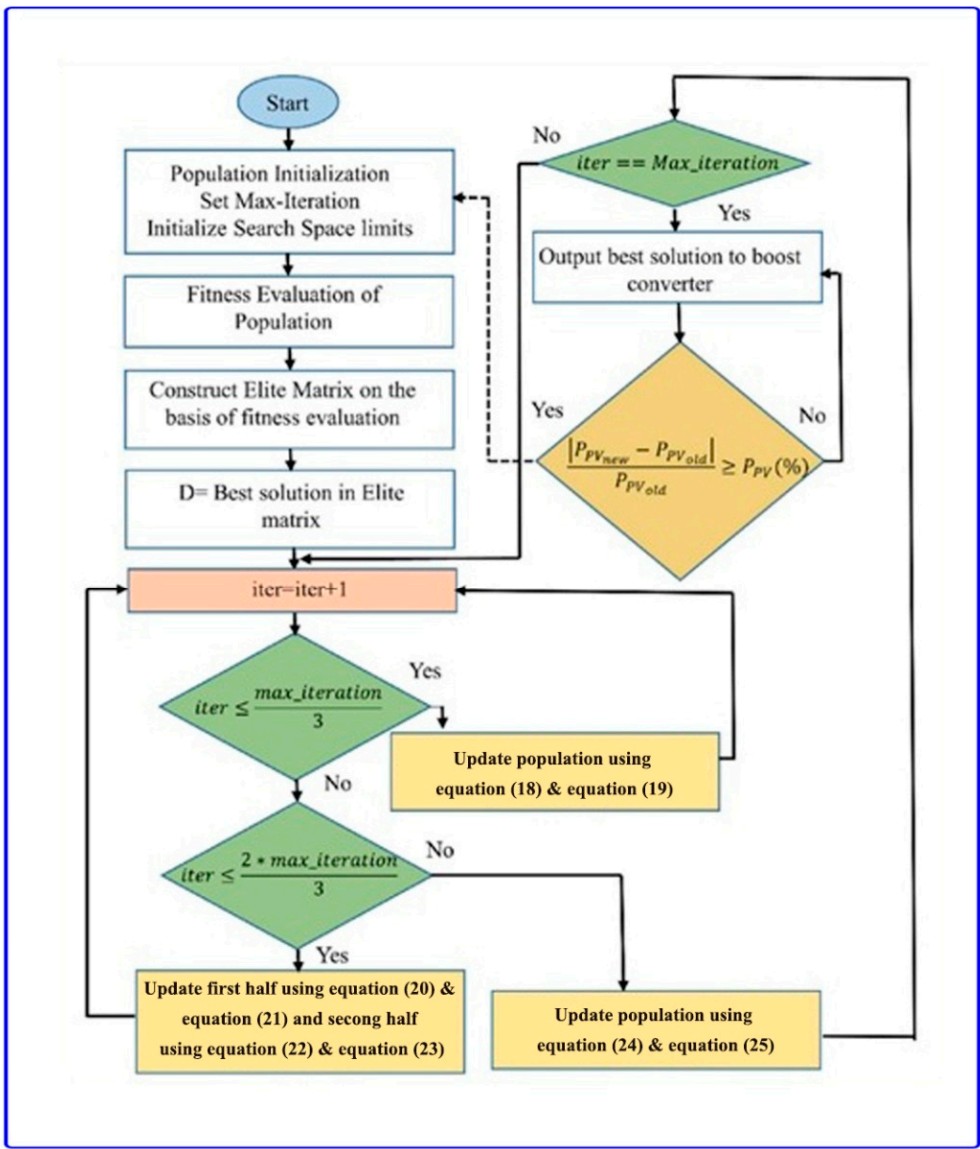

**Figure 10.** Flowchart for marine predator algorithm-based MPPT.

To provide maximum power for the EV, the above-mentioned process using MPA will track and extract the maximum power from the solar PV.

## 5. BLDC Motor Control Based on PID

PID consists of a set of conditions that could be applied to give a closed-loop control system precise regulation. In a closed-loop control process, the controlling device receives continuous real-time measurements of the process being controlled to ensure it reaches the desired range. The computed value, also known as the "process variable", is made by the controlling device to resemble the indented value, also known as the "set point". To complete the desired work effectively, the PID control algorithm is adapted. The most important of these is proportionate control, which measures the error value and creates proportionate changes to lower the error in the control variable. Proportional control is mostly used in many control systems. The PID controller continuously evaluates the difference between the process variable and the set point and makes the necessary corrections. Derivative control monitors the process variable's rate of change and modifies the output variable to take unexpected changes into account.

Each of the three control functions is directed by a user-defined parameter. These characteristics may differ from one control system to another, and as a result, they must be modified for the best control precision. Finding the values of these parameters is known as PID tuning. Although many people think of PID tuning as "black magic", it is always known as a precise mathematical process.

There are numerous ways to accomplish PID tuning, and any of the methods can be used to tune any system. While some PID controlling methods require more devices than others, they typically produce more accurate output with low effort. The fundamental objective of the PID controller is to execute algorithm-based tuning constants. The control engineer delivers the current plant process value and the operator's intended operating value (set point). In most situations, the controller will conduct to bring the process value as close to the set point as is practical. To perform a simple process control loop, PID algorithms will be implemented by the control engineer.

*PID Controlling*

The main goal of the PID controller is to maintain the constant output level so that there is no difference (error) between the process variable (Pv) and the set point (SP). The valve may control the flow of gas to a heater, the water level of the tank, the temperature of a chiller, the flow through a pipe, the pressure of the pipe, or any other method for process control and shown in Figure 11.

$$obj = k_p^* e_t^* dqO + k_i^* \int_0^1 e_t^* dqOdt + k_d^* \frac{de_t^*, dqO}{dt} \tag{28}$$

$$Min(X) = Min(ITAE^*) \tag{29}$$

where, $X$ = total controller error, $ITAE^*$= absolute integral-time error.

$$ITAE^* = \int_0^\infty T|e_t^* dpO|dt \tag{30}$$

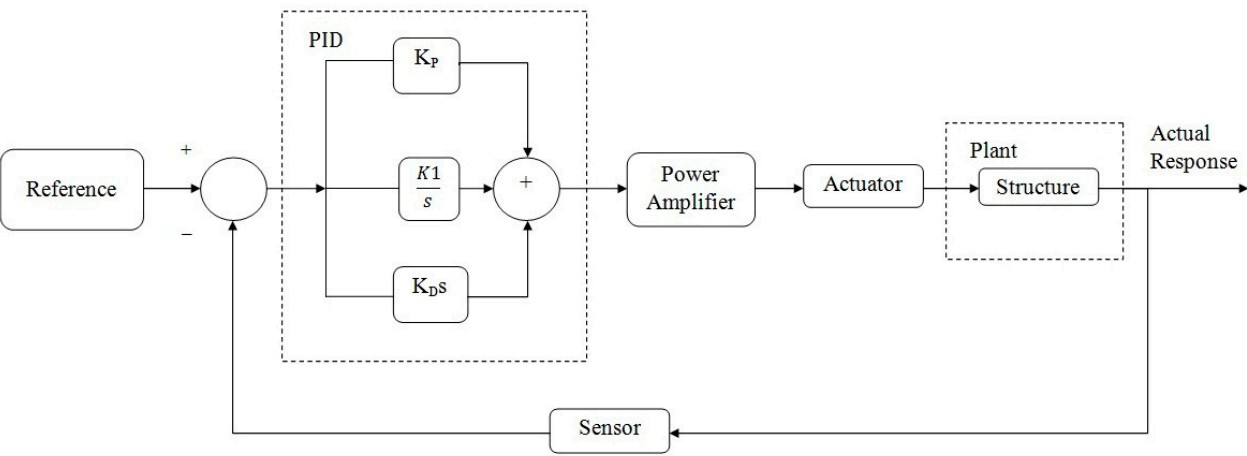

**Figure 11.** PID controller block diagram.

$e_t^* dpO$ = error signal between a reference voltage and load.

The MPA-based control method is implemented in the proposed work. It is explained in Section 4 and provides a detailed explanation of the PID approach utilized for BLDC motors with electronic commutation. It is the most common strategy for controlling the 3ϕ AC motor speed and torque by utilizing the current control method. When the BLDC motors are operated at both high and low speeds, at that time there is no precise speed control and ripple in torque. For applications such as washing machines, PID is very important. The VSI switching signals were produced by the motor's "electronic commutation". The Hall

effect sensor is attached to the stator and is used to find the rotor position angle. These Hall effect signals are changed into six switching pulses, which are used to control the switches of the voltage source inverter.

## 6. Results and Discussion

The solar PV panels used to generate the power of 500 W, in addition to power connecting the DC-DC converter, are included in the proposed work; it was made by using MATLAB/Simulink software. To charge a vehicle battery, the power extract from PV is used. In this configuration, we are powering a BLDC motor rating of 3000 rpm, 256 V, 1 kW, and a BLDC motor with a series batteries rating of 220 V, 90.4348 Ah. The current and voltage extract from the solar array is given as MPPT input, and the switch in the DC-DC converter is activated by PWM signal, enabling the implementation of the MPPT from the solar array. The proposed MPA-based MPPT control algorithm is correlated to the GWO and WOA MPPT control bioinspired algorithms to assess how well it performs.

The findings of this study are examined in four modes, which are described in the following section:

Mode 1: Constant Motor Speed and Constant Irradiance

The MATLAB Simulink software is used to simulate the suggested work under 25 °C constant temperature and a constant irradiation of 1000 W/m$^2$, as shown in Figure 12. Meanwhile the settling time is 0.06 s and 0.035 s for power through the GWO and WOA, respectively. This produces a high fluctuating signal. The output power through the MPA is settled in 0.02 s. In the first case, the PV panel temperature is 25 °C constant, and the PV irradiance is 1000 W/m$^2$. In affixing, the BLDC motor speed is set at 3000 rpm constant, and the battery voltage and PV power outputs are analyzed in Figure 13. The PV current, PV voltage, and PV power are illustrated in Figure 13; the solar PV power reaches 62 kW and settles in 0.02 s, and the PV current and voltage are obtained at 185 A and 340 V, respectively. Figure 13 demonstrates the outputs of the battery, which are the current, SOC, and output voltage of the battery. The battery current and voltage are obtained at 480 A and 360 V, respectively, and battery SOC is reached at 100% in discharging mode.

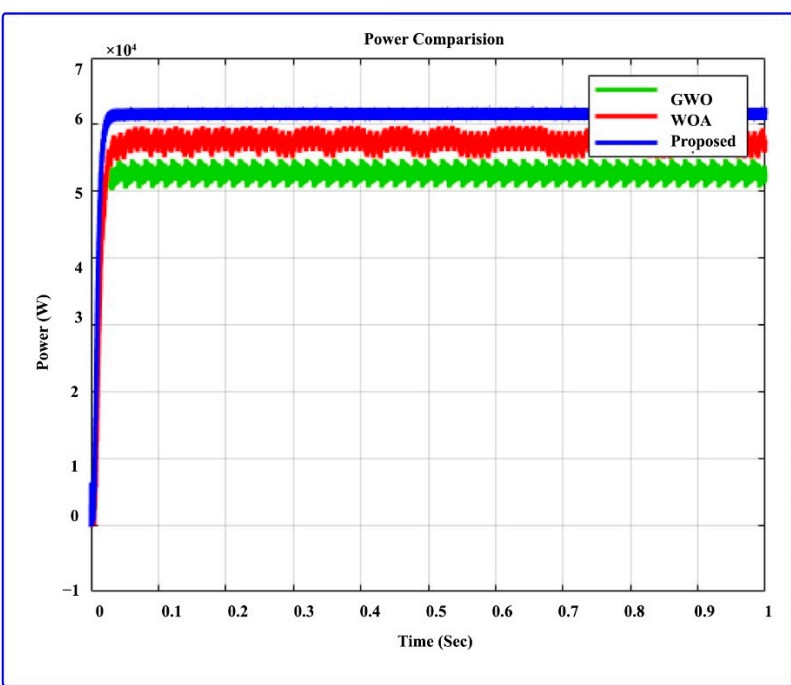

**Figure 12.** PV output power while the solar irradiation and motor speed are both constant.

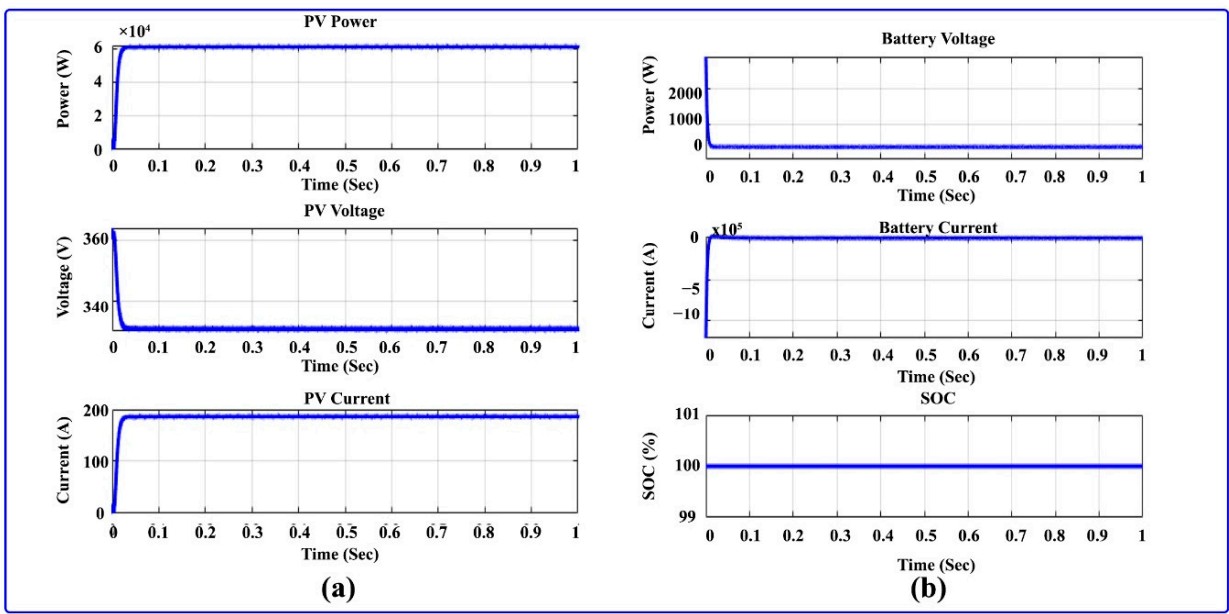

**Figure 13.** (**a**) PV outputs at constant speed and irradiation (**b**) battery outputs at constant speed and irradiation.

## 7. BLDC Motor Outputs

The BLDC motor output is illustrated in Figures 14 and 15. Here, Figure 14 shows the speed and the BLDC motor comparison. At first, 3000 rpm is set as a reference speed for one second. Figure 14a illustrates the BLDC motor speed, which is set to a constant speed of 3000 rpm, and Figure 14b demonstrates the speed and reference speed comparison. Figure 15 shows the stator current and torque of the BLDC motor. Here, a stator current of 3 φ is used to demonstrate how the BLDC motor's torque is reached at 1.2 Nm. Figure 15a at the beginning increases at 100 Nm, and at 0.02 s it decreases to 1.2 Nm. Figure 16 illustrates the Hall signal and back EMF of the BLDC motor.

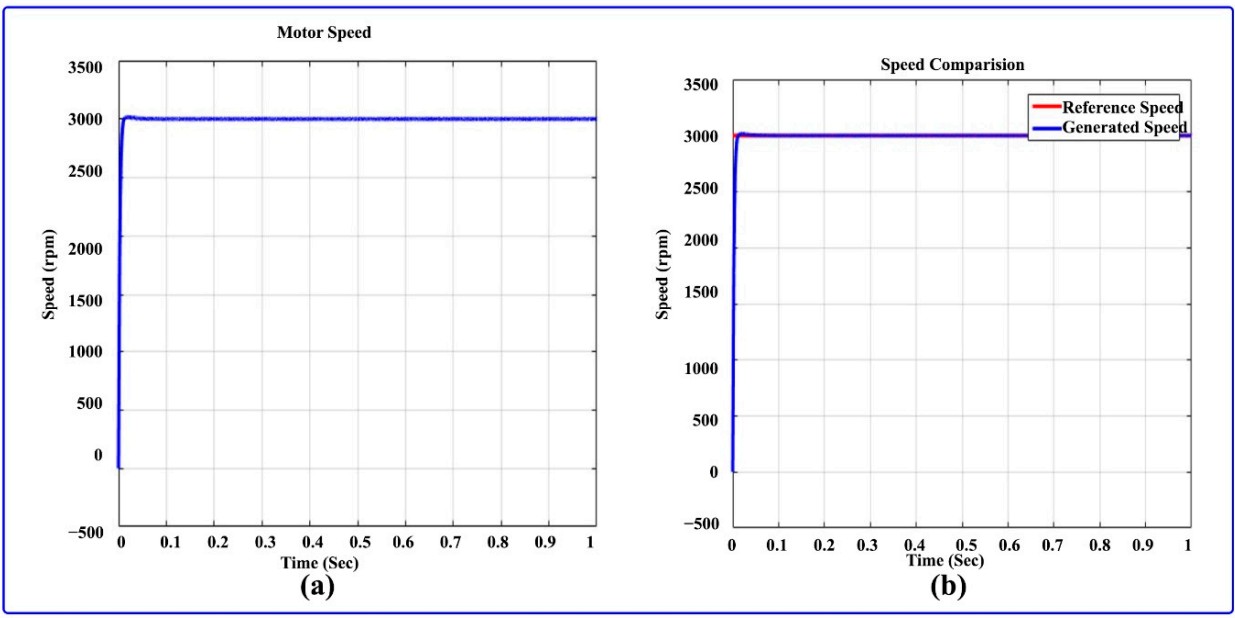

**Figure 14.** (**a**) Speed of the BLDC motor (**b**) comparison of motor speed and reference speed.

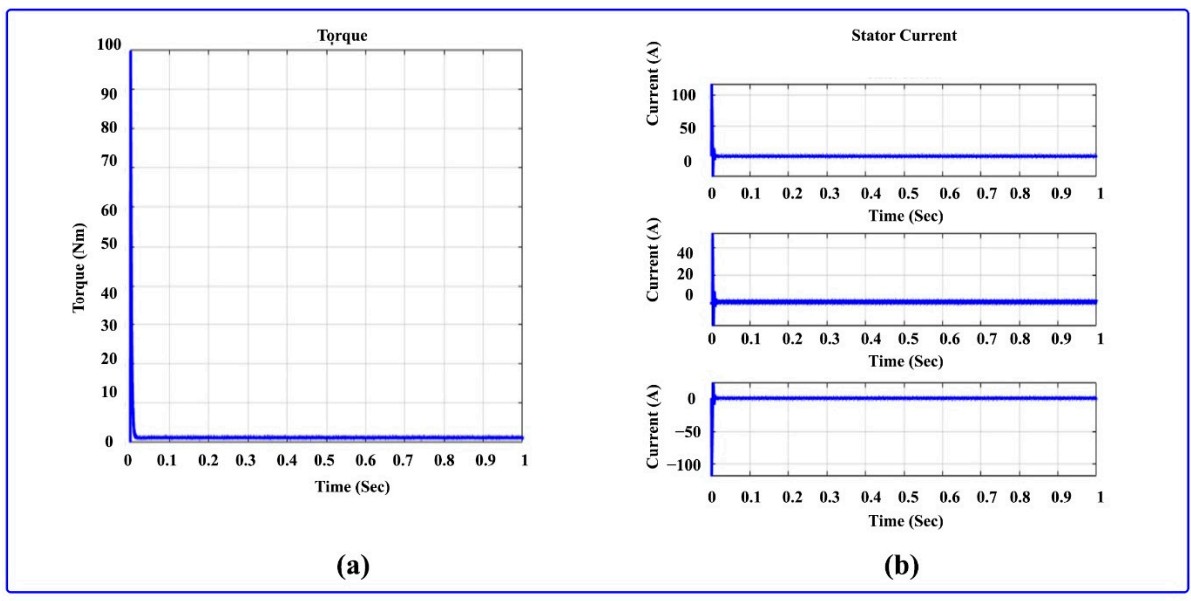

**Figure 15.** (**a**) BLDC motor torque at constant speed (**b**) BLDC motor stator current at constant speed.

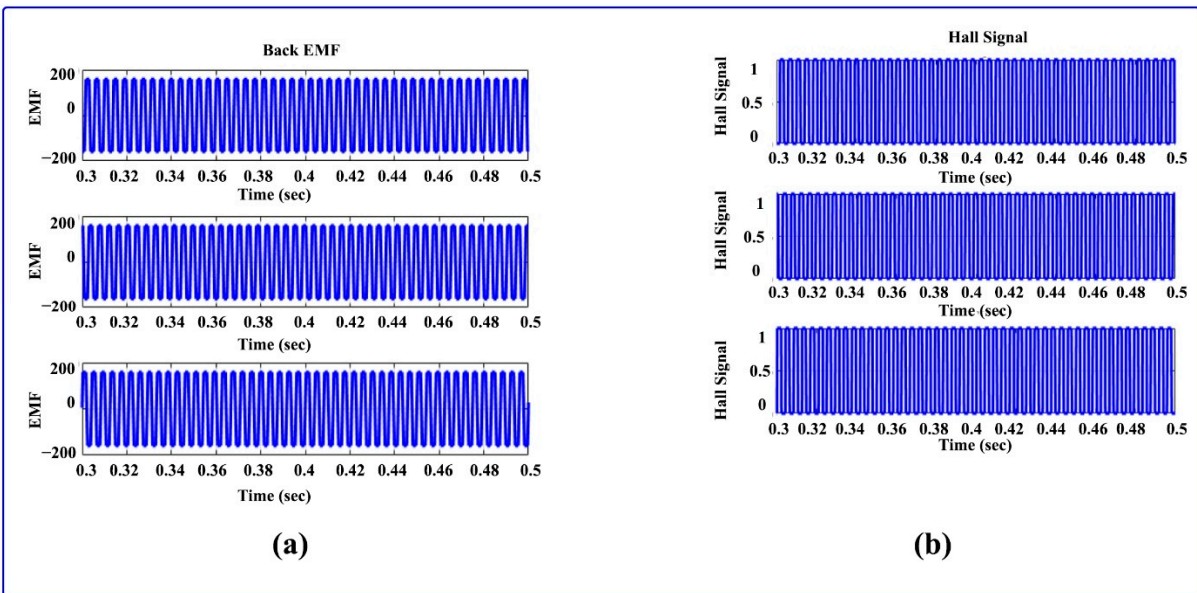

**Figure 16.** (**a**) BLDC motor back EMF at constant speed (**b**) BLDC motor Hall signal at constant speed.

Mode 2: Constant motor speed and varying irradiation

The MATLAB Simulink software is used to simulate the suggested work for 0–0.5 s in 1000 W/m$^2$ irradiation, 0.5–1 s in 500 W/m$^2$, with 3000 rpm motor speed and 25 °C constant temperature. In Figure 17, the power flow across the MPA is settled in 0.02 s. The power through the GWO and WOA is settled in 0.06 and 0.35 s, respectively. This produces a highly fluctuated signal. The suggested power from solar PV reaches 63 kW in 0–0.5 s at an irradiance of 1000 W/m$^2$, as shown in the graph. Then, the power is changed to 40 kW after 0.5–1 s at an irradiance of 500 W/m$^2$. The suggested approach is related to the traditional approach, such as the GWO and WOA MPPT algorithm, and proves the advancement of the suggested method. The battery and PV output is depicted in Figure 18. Figure 18a illustrates the PV power, voltage, and current attained at 62 kW, 340 V, and 185 A, respectively, that were generated in solar PV at 0–0.5 s time range. The irradiance

is 1000 W/m$^2$, and then suddenly PV power, PV current, and PV voltage raise to 61 kW, 175 A, and 325 V, respectively, and irradiance to 500 W/m$^2$.

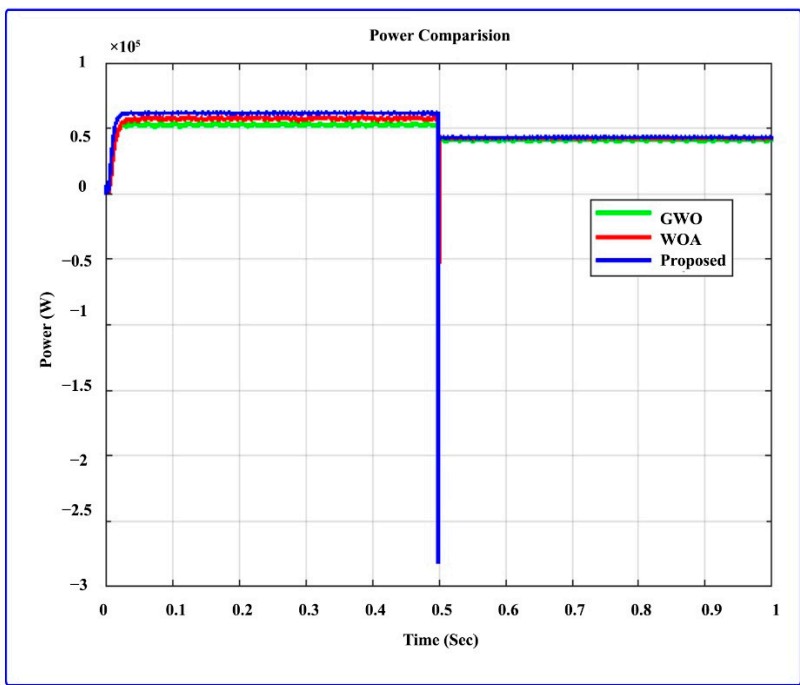

**Figure 17.** PV panel output power with variable irradiance.

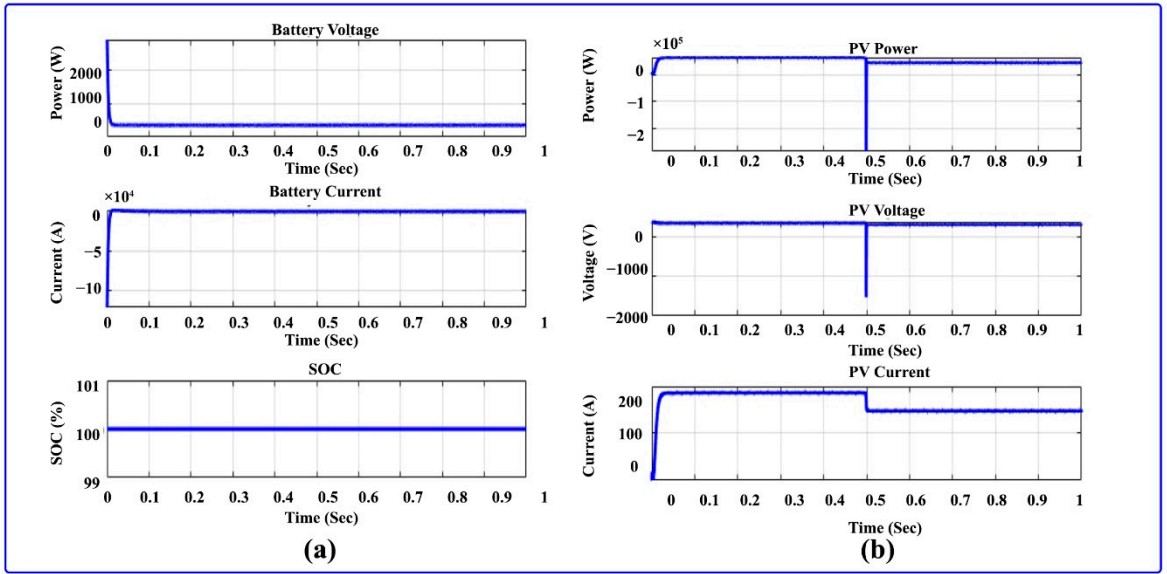

**Figure 18.** (**a**) Battery outputs at variable irradiance (**b**) PV outputs at variable irradiance.

The BLDC motor speed comparison at various irradiances is shown in Figure 19. In Figure 19a, the BLDC motor speed in regard to setting the reference speed of 3000 rpm for 0–0.5 s at PV irradiance is 1000 W/m$^2$. After 0.5 s at the same motor speed, PV irradiance is set to 500 W/m$^2$. The speed of the actual speed-to-reference speed comparisons is depicted in Figure 19b. The torque of the BLDC motor and stator current is shown in Figure 20; Figure 20a shows the high torque starting stage of the motor after 0.01 s, which dropped immediately to settle at 0.01 s on 1.2 Nm. The comparison of speed during the constant speed and variable irradiance of the motor is shown in Figure 20b. The Hall signal and back EMF of all phases are shown in Figure 21a,b.

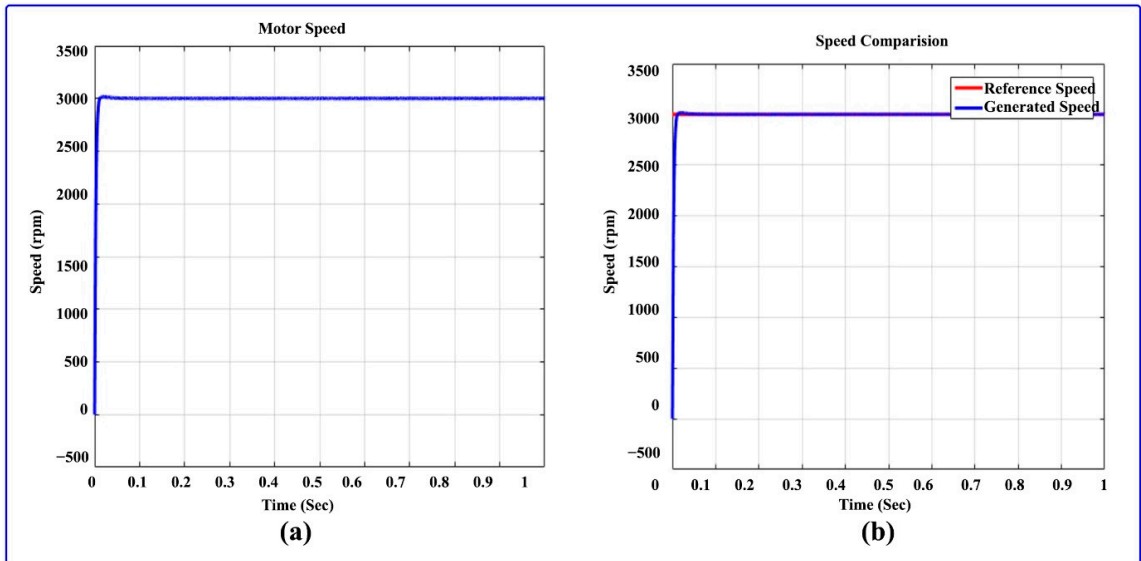

**Figure 19.** (**a**) Speed of the BLDC motor (**b**) comparison of motor speed and reference speed at varying irradiation.

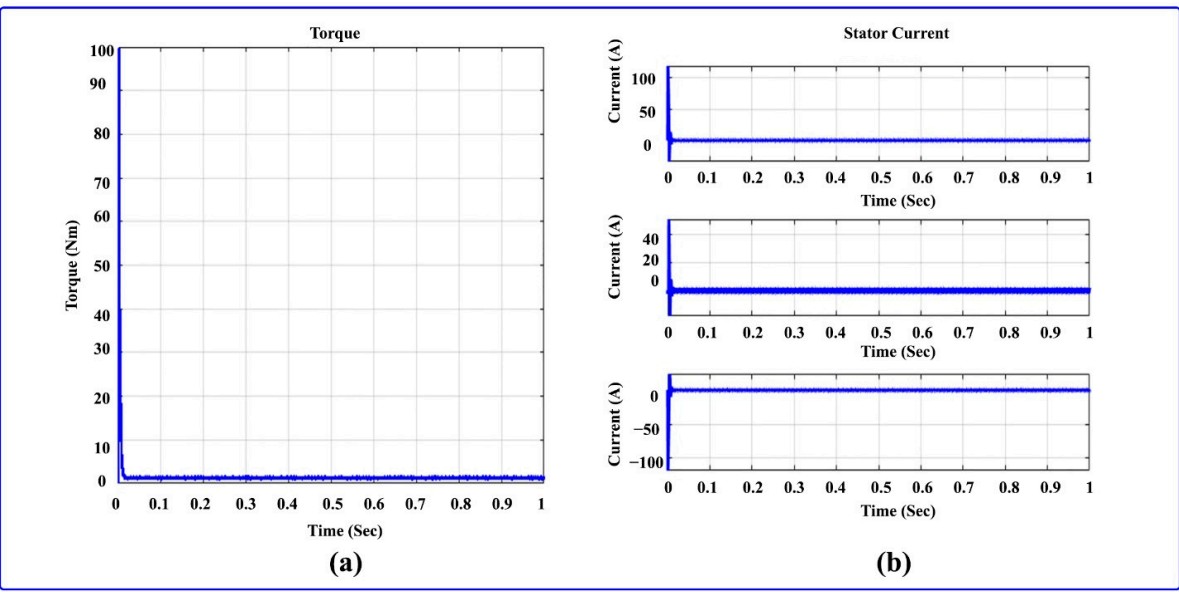

**Figure 20.** (**a**) BLDC motor torque with variable irradiation and constant speed (**b**) BLDC motor stator current with variable irradiation and constant speed.

Mode 3: Variable motor speed and constant irradiation

The input for this mode is variable motor speed and constant irradiance; 1000 W/m$^2$ for constant irradiance, 3000 W/m$^2$ for 0–0.2 s, 1000 W/m$^2$ for 0.2–0.4 s, 1500 W/m$^2$ for 0.6–0.8 s, 2500 W/m$^2$ for 0.6–0.8 s, and 2000 W/m$^2$ for 0.8–1 s for BLDC motor speed, respectively. Figure 22 illustrates the PV power at the variable motor speed and constant irradiance comparison. The suggested approach is more admirable than the GWO and WOA. According to the variation in the motor speed, the power will be varied. When the motor speed is 3000 rpm, the suggested technique power will reach 62 kW. When the motor speed is decreased to 1000 rpm, the suggested technique power is increased from 62 KW to reach a power of 81 kW. Accordingly, for motor speeds of 1500 rpm, 2000 rpm, and 2500 rpm, the power of the suggested approach will be 78 kW, 70 kW, and 68 kW, respectively. Figure 23 depicts the PV output and battery output at 1000 W/m$^2$ of constant

irradiance, and variable speeds of 3000 between 0 and 0.2 s, 1000 from 0.4–0.6 s, 2500 from 0.6–0.8 s, and 2000 from 0.8–1 s are used. The PV current, power, and voltage are shown in Figure 23a. In that PV power for the regular interval of (0.2, 0.4, 0.6, 0.8), high changes may occur for the regular interval. In the meantime, the voltage of PV is drained to 325 V, and the current of PV is increased to 220 A. The battery SOC, battery current, and voltage of the battery are depicted in Figure 24. The changes in battery current and voltage, as well as battery charging and discharging, are caused by the variable motor-speed input. The BLDC motor Hall signal and back EMF at variable speed and constant irradiance are depicted in Figure 25.

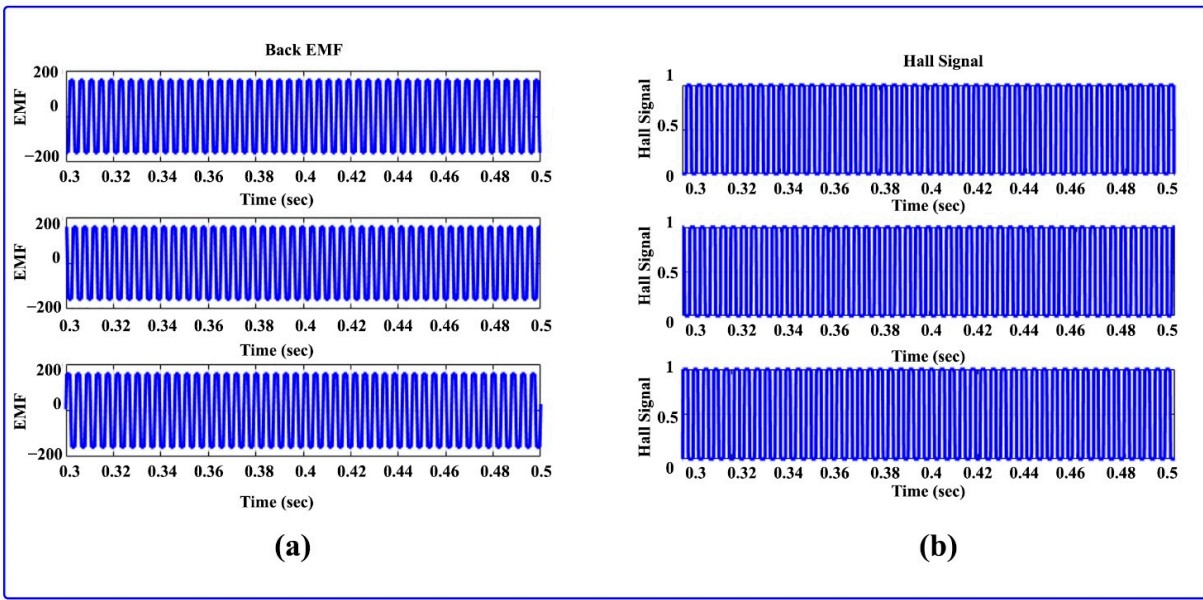

**Figure 21.** (**a**) BLDC motor back EMF with variable irradiation and constant speed (**b**) BLDC motor Hall signal with variable irradiation and constant speed.

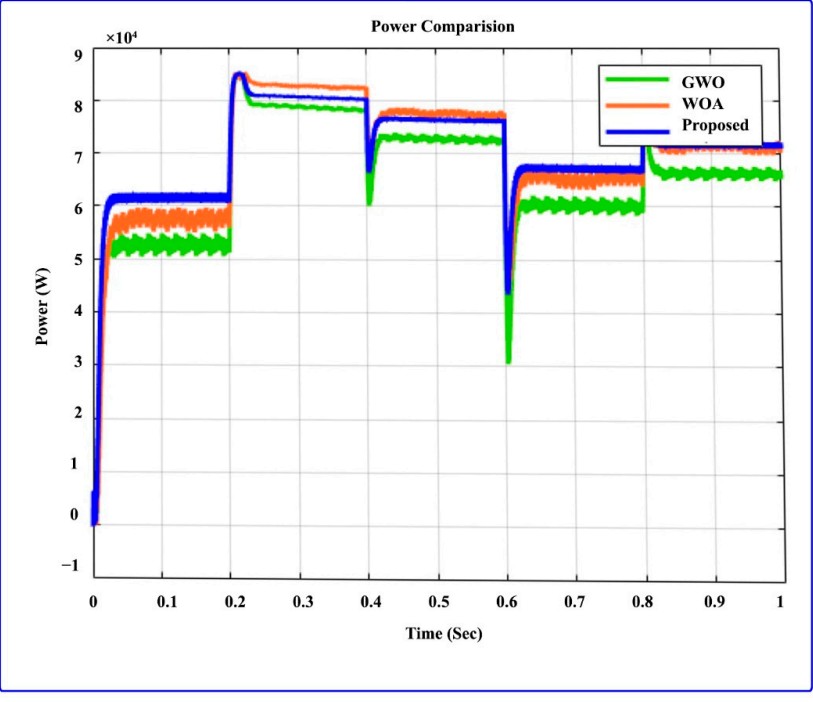

**Figure 22.** PV power comparison with variable motor speed and constant irradiation.

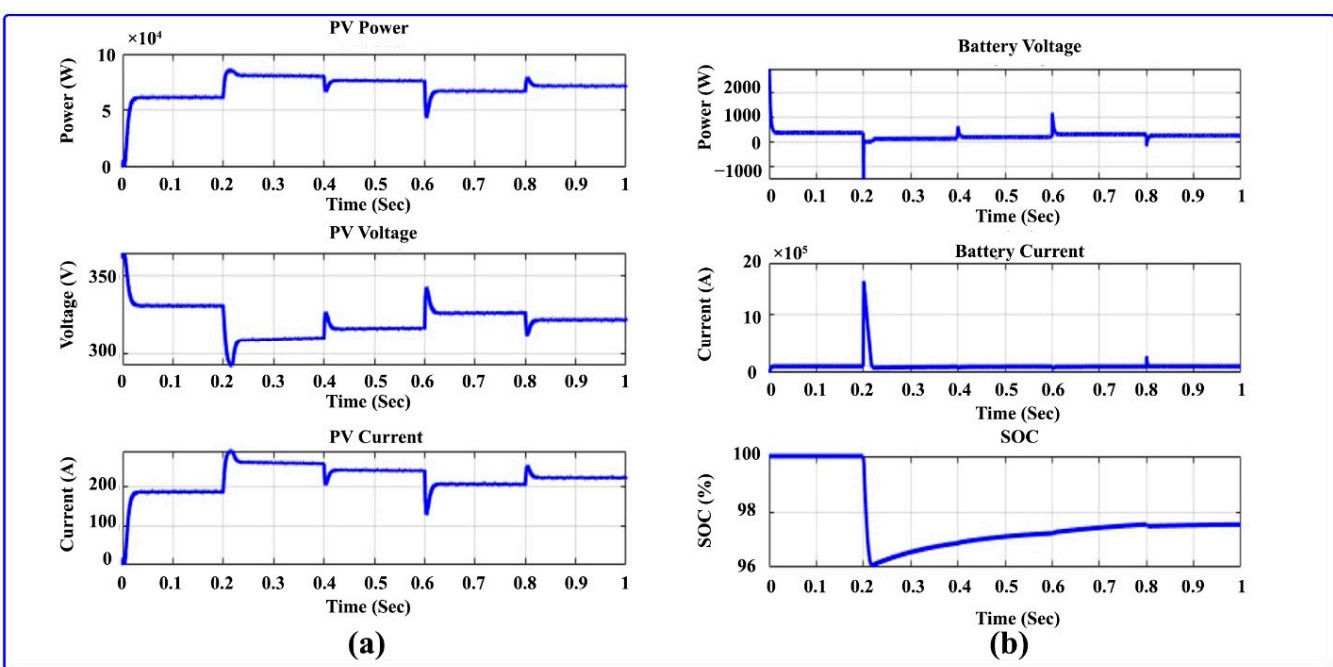

**Figure 23.** (**a**) PV outputs with varying speeds and constant irradiance (**b**) battery outputs with varying speeds and constant irradiance.

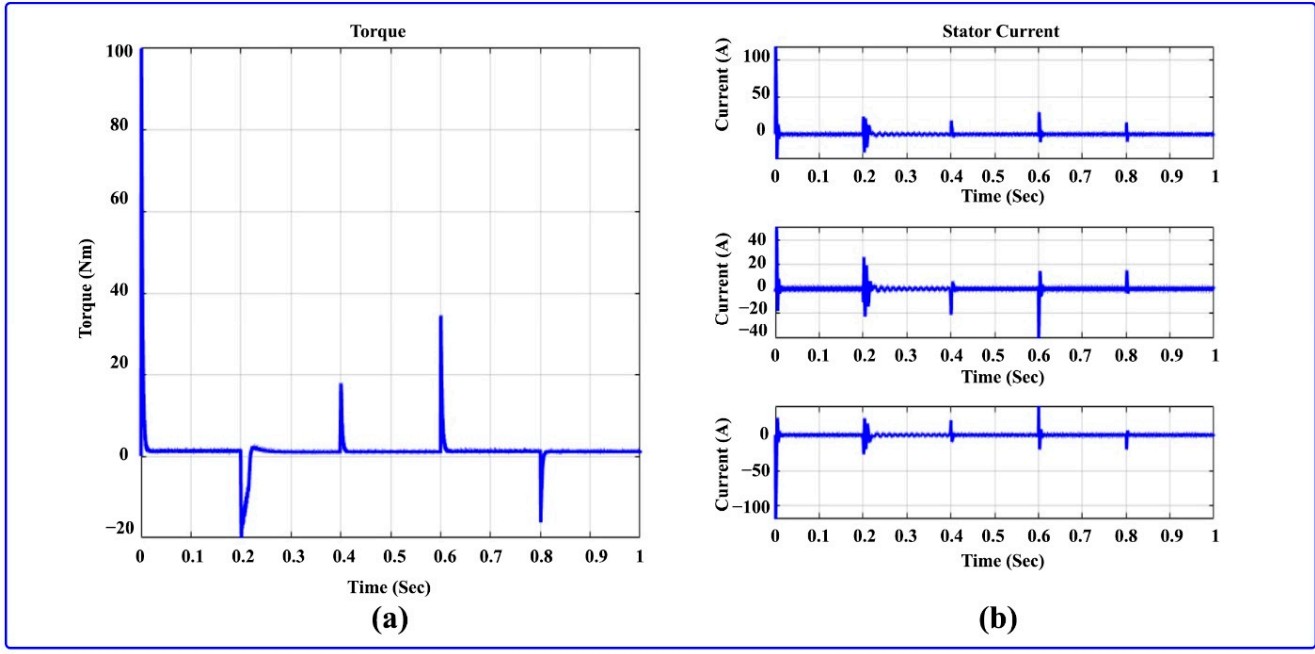

**Figure 24.** (**a**) BLDC motor torque at varying speeds and constant irradiation (**b**) stator current at varying speeds and constant irradiation..

Mode 4: Variable Motor Speed and Variable Irradiance

Figure 26 illustrates the comparison of power output at variable motor speed and variable irradiance. In this mode, variable irradiance is 1000 W/m$^2$ for 0–0.5 s before shifting to 500 W/m$^2$ for 0.5–1 s. Additionally, the speed is fixed at 3000 rpm for 0–0.2 s, 1000 rpm for 0.4–0.6 s, 1500 rpm for 0.6–0.8 s 2500 rpm, and 2000 rpm for 0.8–1 s. The advanced approach is related to the traditional approach to prove its advantages. The PV and battery output are depicted in Figure 27. The output of the solar PV at variable

speed and irradiance is illustrated in Figure 27a. Power from PV varies concerning varying irradiance and variable speed. Between 0 and 0.5 s, the variable irradiance is changed to 1000 W/m$^2$ and 500 W/m$^2$ between 0.5–1 s. Furthermore, the BLDC motor speed is modified, from 0 to 0.2 s, 3000 rpm, 1000 to 0.4 s, 1500 to 0.6 s, 2500 to 0.8 s, and 2000 to 1 s. All outcomes from PV are changed in favor of the variable speed and variable irradiance. Figure 27b shows the outputs of the battery current, battery voltage, and battery SoC. The BLDC motor speed and a comparison of its speed under different irradiation conditions are illustrated in Figure 28. The motor's speed relative to the reference speed for 1000, 1500, 2000, 2500, and 3000 rpm is demonstrated in Figure 28a. The comparison speed to a reference speed is illustrated in Figure 28b. The reference speed is denoted as a straight red line, and the BLDC motor's actual speed is mentioned as a blue line. The variable speed is given to 0, 0.2, 0.4, 0.6, and 0.8 s time intervals at 3000, 1000, 1500, 2500, and 2000 rpm. Figure 29 depicts the BLDC motor stator current and torque. Figure 29a illustrates the torque. Due to modifications made to the speed of the BLDC motor and irradiance of the solar PV panel, the peak and dip on the torque are now visible. Figure 29b illustrates all three-phase stator currents. Solar energy is used by the boost converter to power the 3000 rpm, 48 V, 1 kW BLDC motor and for battery charging. The operation of a variable speed BLDC motor is shown in Figure 28. We started the motor for 0.2 s and set the reference speed of 3000 rpm. After that, for 0.4 s it is shifted to 1000 rpm, then for 0.6 s it is changed to 1500 rpm, and at the end, it is changed to 2500 rpm. The speed of the motor reaches the designated reference speed in less than 0.01 s. Figure 29a shows the variation in the torque, first supplied at 100 Nm for 0.01 s. Due to speed changes, there are a few spikes in the torque. Figure 29b illustrates the motor current, the result of which is a spike representing a change in speed. Figure 30 illustrates the BLDC motor Hall signal and BLDC motor back EMF; it is varied with variation in speed of 1000, 1500, 2000, 2500, and 3000 rpm. According to the above results, the proposed approach is superior to the existing approach.

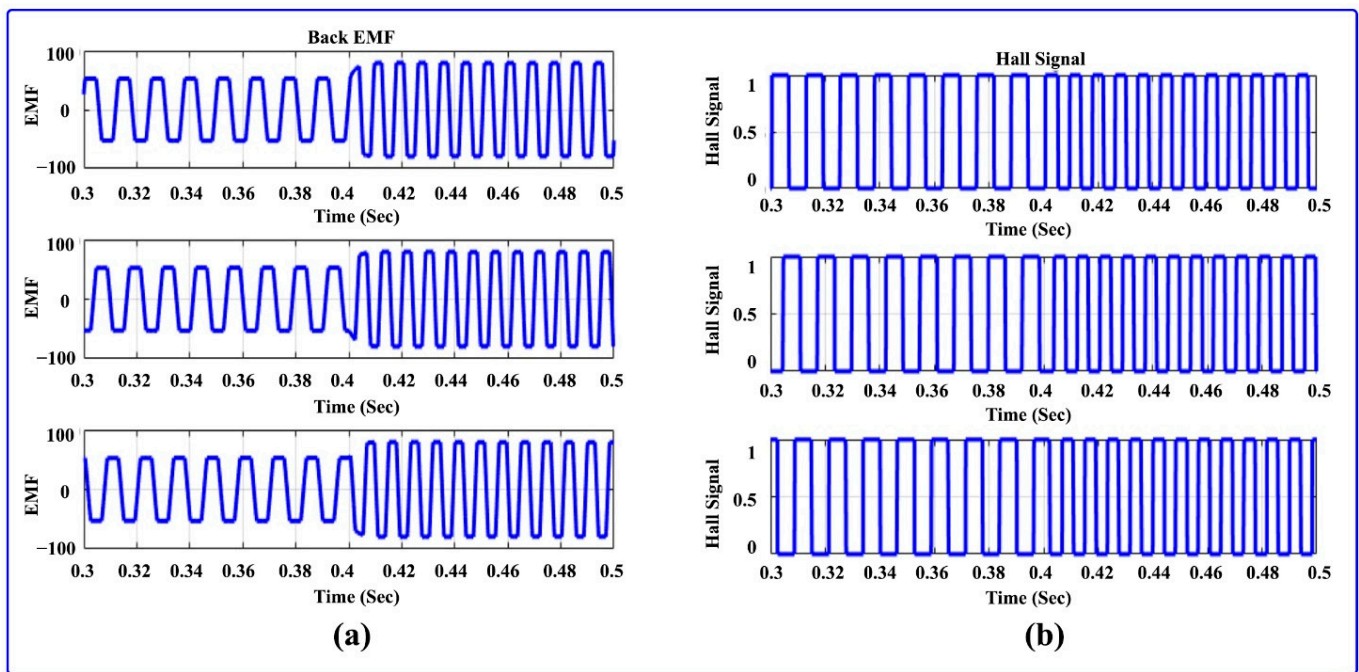

**Figure 25.** (**a**) Back EMF of BLDC motor (**b**) Hall signal of BLDC motor operating at variable speed and constant irradiation.

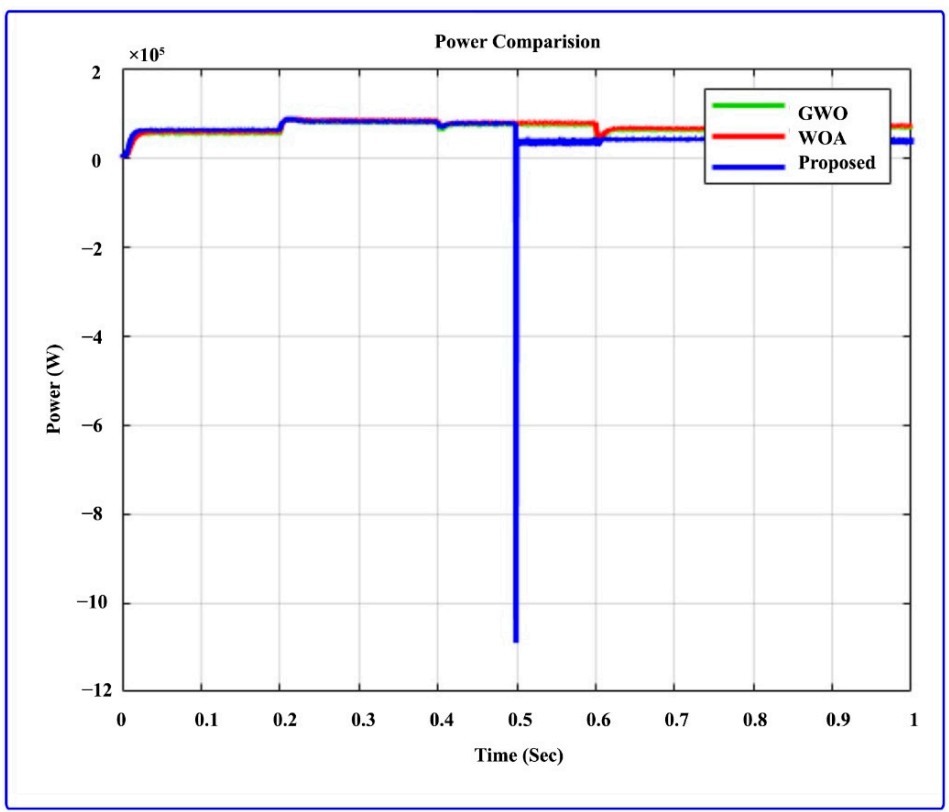

**Figure 26.** Comparison of output power with varying irradiation and varying motor speed.

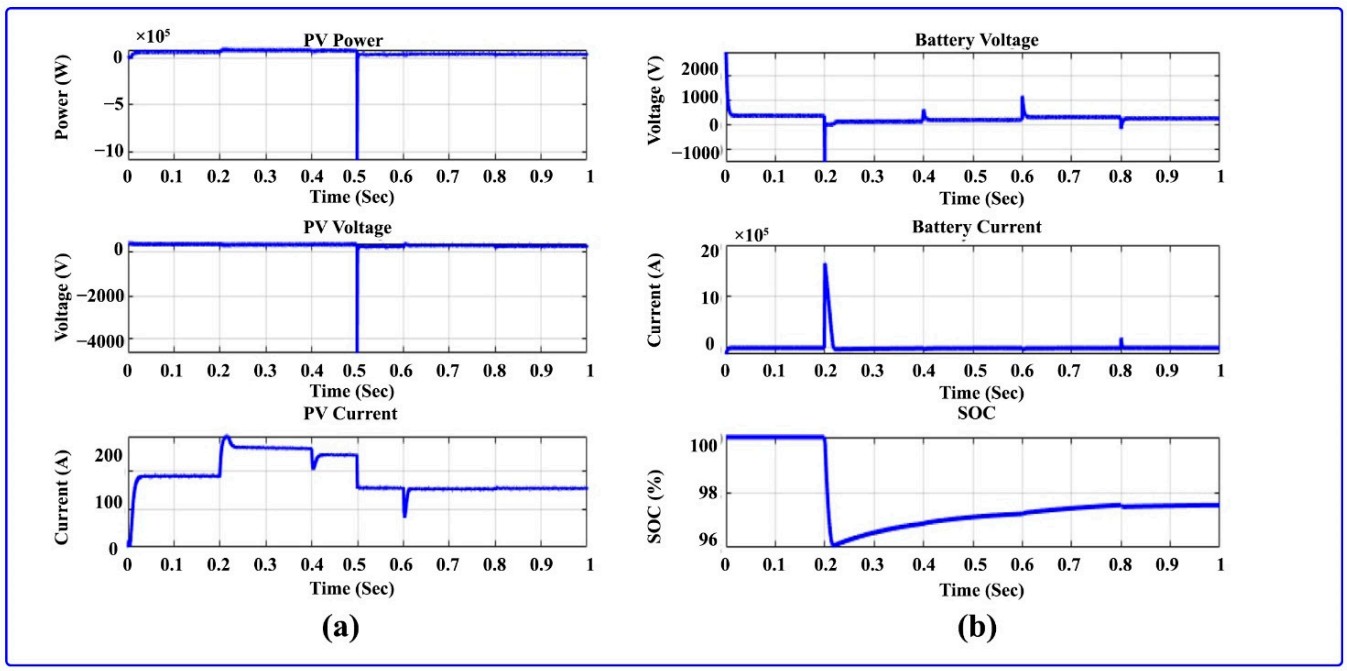

**Figure 27.** (**a**) PV outputs with varying irradiance and varying speed (**b**) battery outputs with varying irradiance and varying speed.

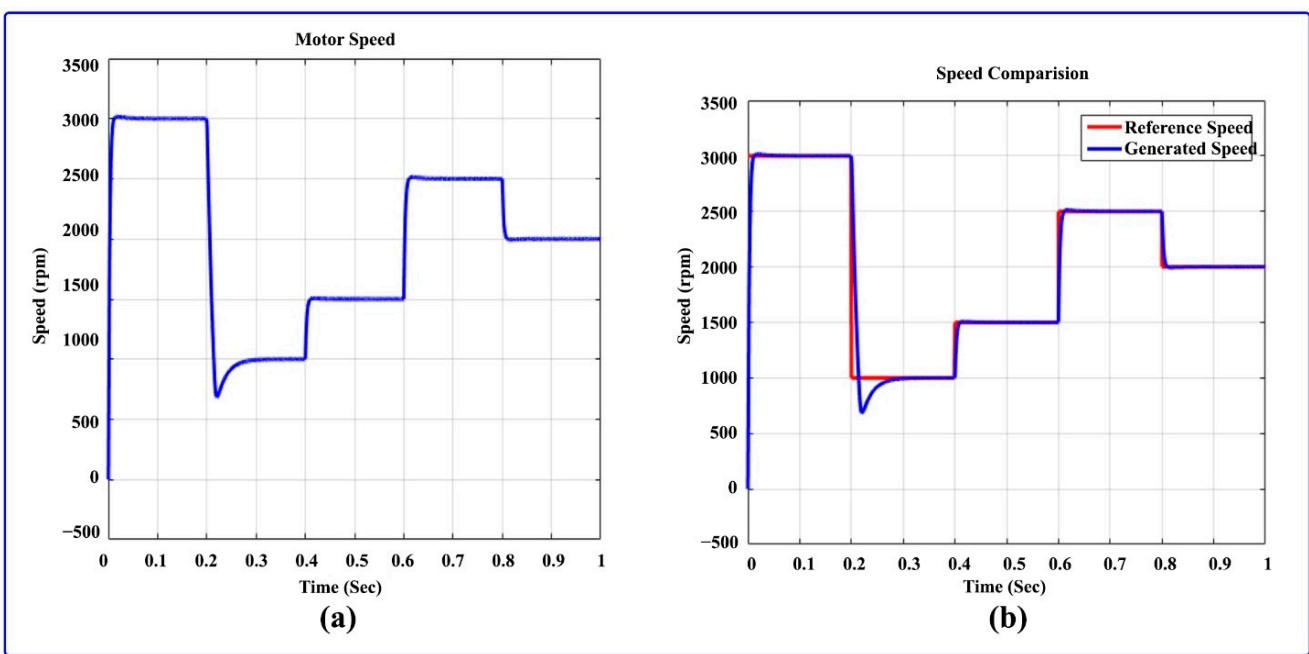

**Figure 28.** (**a**) BLDC motor speed (**b**) comparison of motor speed and reference speed with variable irradiance and variable speed.

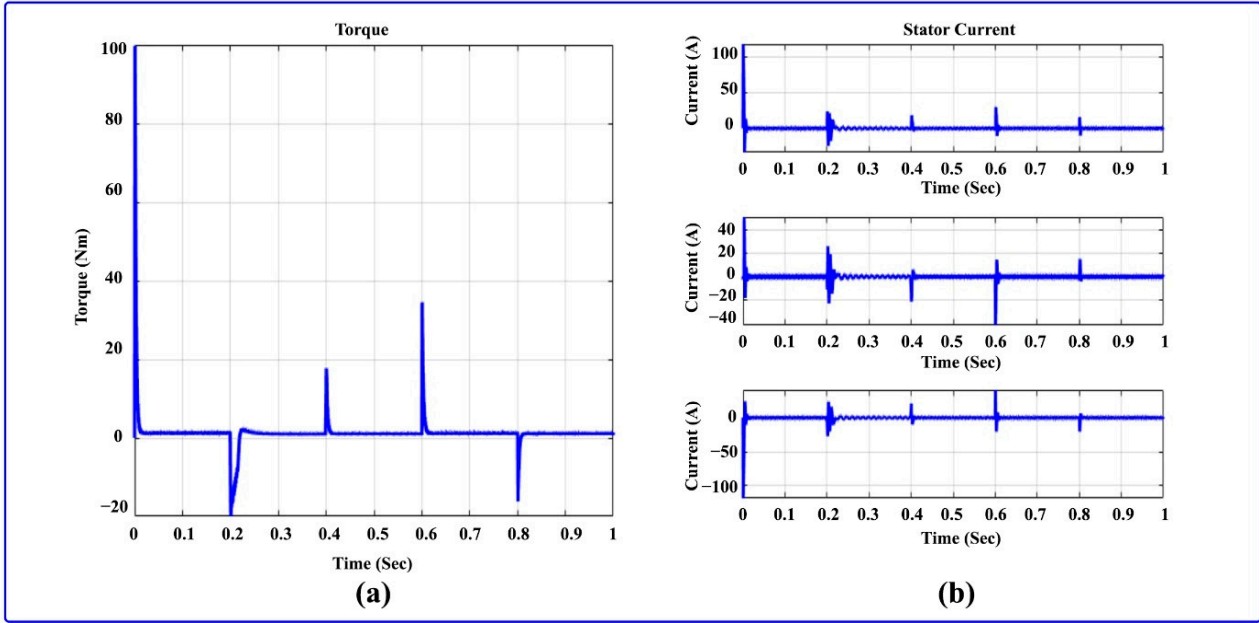

**Figure 29.** (**a**) BLDC motor torque with varying speed and irradiation (**b**) stator current with varying speed and irradiation.

Tables 1 and 2 present the qualitative and quantitative comparative analysis of the existing [29] and proposed optimization-based MPPT techniques, respectively. During quantitative analysis, the tracking time and efficiency measures are compared with the conventional P&O, FLC, and AFLC mechanisms. Then, the overall performance of the MPPT controlling algorithms is validated and compared during qualitative analysis based on the parameters of tracking speed, complexity, tracking efficiency, reliability, MPP oscillations, and tracking accuracy. The MPA-MPPT controlling technique provides highly improved results compared to the standard MPPT techniques.

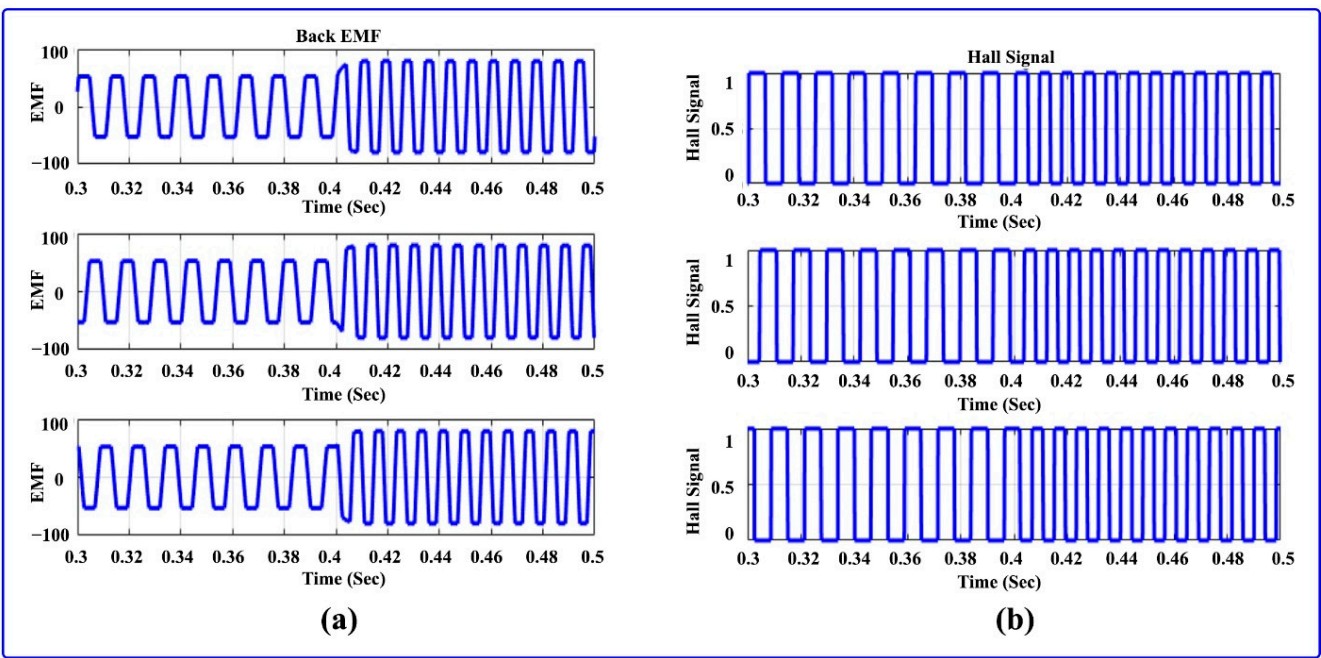

**Figure 30.** (**a**) Back EMF for BLDC motor with varying speed and irradiation (**b**) Hall signal for BLDC motor with varying speed and irradiation.

**Table 1.** Quantitative analysis.

| Methods | Tracking Time | Efficiency |
|---------|---------------|------------|
| P&O | 0.05 | 99.94 |
| FLC | 0.05 | 99.96 |
| AFLC | 0.038 | 99.97 |
| Proposed MPA | 0.025 | 99.98 |

**Table 2.** Qualitative analysis.

| Criteria | P&O | FLC | ACO-FLC | Fuzzy-PSO | GWO-FLC | Proposed MPA |
|----------|-----|-----|---------|-----------|---------|--------------|
| Tracking Speed | Slow | Moderate | Moderate | Moderate | Fast | Very Fast |
| Complexity | Less | Less | Moderate | Moderate | Less | Very Less |
| Tracking Efficiency | Less | Less | Medium | Medium | High | Very High |
| Reliability | Low | Low | Low | High | High | Very High |
| MPP Oscillations | High | High | Moderate | High | Less | Very Less |
| Tracking accuracy | Medium | Medium | Medium | Medium | Accurate | High Accurate |

Table 3 presents the comparative analysis of existing [30] and proposed optimization-based MPPT controlling techniques based on the parameters of convergence time, settling time, and efficiency. Then, its corresponding graphical illustrations are presented in Figures 31 and 32, respectively. The estimated analysis proves that the time of the proposed MPA technique is greatly increased with high efficiency, which is highly superior to the other MPPT controlling techniques.

**Table 3.** Comparative analysis.

| Methods | Convergence Time (s) | Settling Time (s) | Efficiency (%) |
|---|---|---|---|
| SRA | 0.1811 | 0.2402 | 99.98 |
| GHO | 0.3112 | 0.5621 | 99.89 |
| GWO | 0.4421 | 0.6514 | 99.88 |
| PSOGS | 0.3522 | 0.6112 | 99.91 |
| CS | 0.3801 | 0.7701 | 99.84 |
| PSO | 0.4501 | 0.7102 | 99.86 |
| **Proposed MPA** | **0.1532** | **0.2187** | **99.99** |

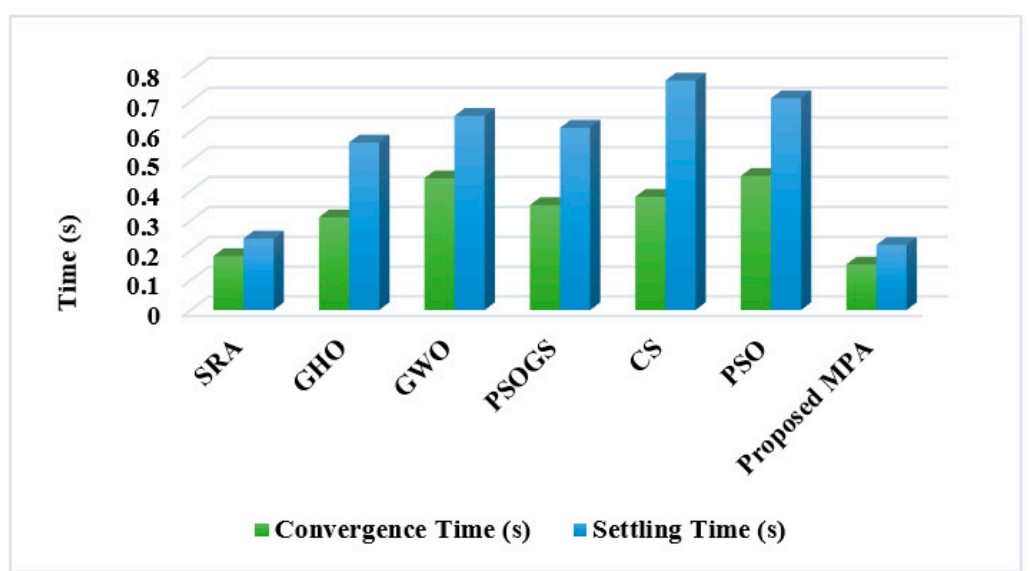

**Figure 31.** Time analysis.

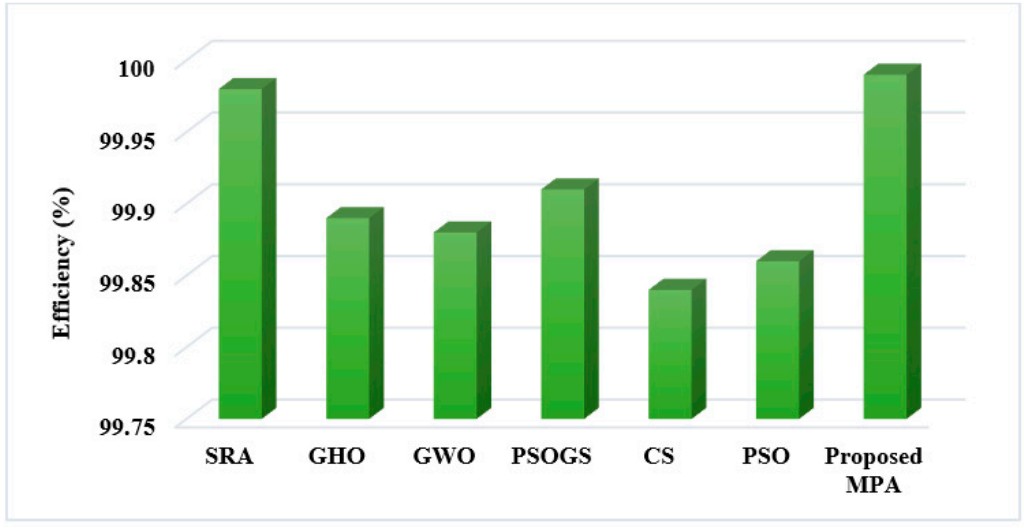

**Figure 32.** Efficiency analysis.

## 8. Conclusions

This research proposes an advanced metaheuristic MPA optimization approach used in a "small electric vehicle system", which operates on a "solar-powered BLDC motor" system. An MPA optimization approach is implemented to retain "Maximum Power Point"

tracking from partial shadow condition as well as constant irradiation in the PV cell in this case. The traditional "MPPT algorithms", especially "WOA" and "GWO", depending on the results of MPPT, were examined, and the proposed algorithm was compared to them. To charge the battery, PV energy is used, and it is also used to supply power to the BLDC motor. The proposed system is created by using MATLAB software. Through the use of torque and change in speed accelerating and decelerating, in addition to the PID controller, the BLDC motor's initial, dynamic, and steady-state behaviors were evaluated. According to the simulation result, the MPA optimization technique improves the performance of the motor and charges the battery well. Consequently, due to continuous solar charging throughout the daytime, the battery is used to operate the BLDC motor for more distance than any electric vehicle.

**Author Contributions:** Data curation: R.K.G.R. and P.K.B.; Writing original draft: R.K.G.R.; Supervision: T.S., P.K.B. and U.M.; Project administration: P.K.B., U.M. and T.S.; Conceptualization: U.M. and A.M.M.S.; Methodology: P.K.B. and R.K.G.R.; Validation: A.M.M.S. and U.M.; Visualization: U.M. and A.M.M.S.; Resources: T.S. and P.K.B.; Review & Editing: T.S., P.K.B. and R.K.G.R.; Funding acquisition: T.S. All authors have read and agreed to the published version of the manuscript.

**Funding:** This research received no external funding.

**Institutional Review Board Statement:** Not applicable.

**Informed Consent Statement:** Not applicable.

**Data Availability Statement:** Not applicable.

**Conflicts of Interest:** The authors declare no conflict of interest.

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
