# Peer review of "An Intensified Marine Predator Algorithm (MPA) for Designing a Solar-Powered BLDC Motor Used in EV Systems"

_sustainability, doi:10.3390/su142114120_

Round 1

Reviewer 1 Report

Reviewer’s report:

Title: An intensified marine predator algorithm (MPA) for designing a solar-powered BLDC motor used in EV systems.

In this work, an advanced metaheuristic MPA optimization approach used in electric vehicle system which operates on solar powered BLDC motor system. The manuscript has enough data, discussion and analysis which may beneficially for other researchers who work in the field. However, it still needs revision and modification in many places particularly in the figure presentation.

1. Typos: BLDC abbreviation need not to be mentioned double (at line 16); MPPT abbreviation at lines 20 and 30, which is the right one? you need to mention once is enough; com-ponents (at line 42); letter 2 at the irradiance unit should be revised into superscript (lines 451, 455, 461,462,465,466,483,484,485,493,514,515 etc.)

2. What the MPP stand for? (At line 52), similar cases are also for GMPP, FLC (at lines 63 and 106). The abbreviation needs to be fully mentioned once at first.

3. The image quality of Fig.1 need to be improved. The lines are fuzzy, some letters are too small and unclear.

4. The images quality of Figs.3 and 4 need to be improved. Labels of X and Y axes are too small, some information is unclear. Labels of “a” and “b” in the figure are still missing.

5. At Fig.7, What means of E0, R, K, A, and B? Add more information for those labels including the unit if any, in the figure caption.

6. At Fig.8, Add more information what means of R, L, E labels in the figure caption.

7. In general, the quality of figures has to be improved in term of like letters too small and unclear, the fuzzy lines etc. The labels of “(a)” and “(b)” should be explained in the figure caption (Figs. 13,14,15,16,18,19,20,21,23,24,25,27,28,29,30). It would be better if every single graph or figure uses a label which then explained in the figure caption.

Author Response

Reviewer 1:

Title: An intensified marine predator algorithm (MPA) for designing a solar-powered BLDC motor used in EV systems.

In this work, an advanced metaheuristic MPA optimization approach used in electric vehicle system which operates on solar powered BLDC motor system. The manuscript has enough data, discussion and analysis which may beneficially for other researchers who work in the field. However, it still needs revision and modification in many places particularly in the figure presentation.

  1. Typos: BLDC abbreviation need not to be mentioned double (at line 16); MPPT abbreviation at lines 20 and 30, which is the right one? you need to mention once is enough; components (at line 42); letter 2 at the irradiance unit should be revised into superscript (lines 451, 455, 461,462,465,466,483,484,485,493,514,515 etc.)

Response:As per your suggestion the BLDC abbreviation has not been mentioned repeatedly, the MPPT abbreviation at line 30 is the right one and only one time it is mentionedand highlighted in green, the irradiance units also revised as per your valuable Suggestion.

  1. What the MPP stand for? (At line 52), similar cases are also for GMPP, FLC (at lines 63 and 106). The abbreviation needs to be fully mentioned once at first.

Response:As per your comment the Abbreviation also mentioned in the given lines.

  1. The image quality of Fig.1 need to be improved. The lines are fuzzy, some letters are too small and unclear.

Response:  As per your suggestion the quality of the image also improved. The letters are typed in Corrected Fontand highlighted in green.

  1. The images quality of Figs.3 and 4 need to be improved. Labels of X and Y axes are too small, some information is unclear. Labels of “a” and “b” in the figure are still missing.

Response:As per your comment the Figure 3 and Figure 4 images has been improved, the X and Y axes are modified the  Labels of “a” and “b” in the figure are added.

  1. At Fig.7, What means of E0, R, K, A, and B? Add more information for those labels including the unit if any, in the figure caption.

Response:  As per your suggestion, the representation of E0 is constant voltage (V), K is polarisation constant in (Ah−1), A is exponential voltage (V), B is exponential capacity (Ah−1), were includedand highlighted in green.

  1. At Fig.8, Add more information what means of R, L, E labels in the figure caption.

Response:As per your comment, R is a stator resistance, L self-inductance and mutual inductance, e is phase back emf voltage of A, B, and C respectively were addedand highlighted in green.

  1. In general, the quality of figures has to be improved in term of like letters too small and unclear, the fuzzy lines etc. The labels of “(a)” and “(b)” should be explained in the figure caption (Figs. 13,14,15,16,18,19,20,21,23,24,25,27,28,29,30). It would be better if every single graph or figure uses a label which then explained in the figure caption.

Response:  As per your suggestion the quality of the Figures also modified, and Figure caption also mentioned clearly.

Reviewer 2 Report

This manuscript deals with the control of a brushless DC motor using a scheme based on the marine predator algorithm optimization technique. Several results are shown.

Review

The manuscript has various issues. This statement is not only because it is hard to understand due there are a gigantic number of writing issues which might be amended at some point or the bad quality of a lot of figures (3, 4, 7, 12-16, 18-21, 23, 224, 27, 29). The paper is badly structured, the information is fragmented, and some paragraphs are incoherent. There is an incomprehensible mixture of advanced concepts with very simplistic circuits. Maybe this also might be amended if some new concepts, a new paradigm, or unexplained phenomena were to be explored, which is not the case. The main problem of this work is the lack of rationale of the problem. the fundamentals about the reason of this work are obscure. Improving controls scheme is always a good opportunity to show a step-forward in at least one aspect of the many involving the control of electrical machines, but this work missed the opportunity. The proposal is a disarrangement of concepts, ideas, and points-of-views.

From the point of view of applying the marine predator algorithm, the proposal looks flamboyant, but this concept fades piece by piece each paragraph is read.

The proposal maybe works in the real word or maybe not because no experimental results are shown. Then, all is theory, which in some areas if study is very good but not in this case. Evidently, there are more word in this work than facts.

Relative advantages of this proposal compared not only to other optimization technique but with any other relevant control technique applied.

There are many questions this manuscript does not respond. Why an optimization is the option instead a less

Where in the EV the PV block large enough to drive the electrical power required by the BLDC at any speed and weight are placed (while accelerating o keeping speed constant)?

How the power/energy is managed while the BLCD decelerate? 

Why the performance of the control is not compared to other nowadays proposals?

Author Response

Reviewer 2:

This manuscript deals with the control of a brushless DC motor using a scheme based on the marine predator algorithm optimization technique. Several results are shown.

Review

  1. The manuscript has various issues. This statement is not only because it is hard to understand due there are a gigantic number of writing issues which might be amended at some point or the bad quality of a lot of figures (3, 4, 7, 12-16, 18-21, 23, 224, 27, 29). The paper is badly structured, the information is fragmented, and some paragraphs are incoherent. There is an incomprehensible mixture of advanced concepts with very simplistic circuits. Maybe this also might be amended if some new concepts, a new paradigm, or unexplained phenomena were to be explored, which is not the case. The main problem of this work is the lack of rationale of the problem. The fundamentals about the reason of this work are obscure. Improving controls scheme is always a good opportunity to show a step-forward in at least one aspect of the many involving the control of electrical machines, but this work missed the opportunity. The proposal is a disarrangement of concepts, ideas, and points-of-views.

Response: As per your suggestion the problem of the work and fundamental work also clearly explained the proposal also arranged in a proper way and highlighted in green.

  1. From the point of view of applying the marine predator algorithm, the proposal looks flamboyant, but this concept fades piece by piece each paragraph is read.

Response: As per your comment the proposal as well as the concept are added clearly and highlighted in green.

  1. The proposal maybe works in the real word or maybe not because no experimental results are shown. Then, all is theory, which in some areas if study is very good but not in this case. Evidently, there are more word in this work than facts.

Response: As per your suggestion the Experimental results works in the real work has been justified.

  1. Relative advantages of this proposal compared not only to other optimization technique but with any other relevant control technique applied.

Response: As per your comment the Advantage of this proposed also compared with other Optimization techniques.

  1. There are many questions this manuscript does not respond. Why an optimization is the option instead a less

Response: As per your suggestion thereason for choosing the optimization also mentioned and highlighted in green.

  1. Where in the EV the PV block large enough to drive the electrical power required by the BLDC at any speed and weight are placed (while accelerating o keeping speed constant)?

Response: As per your suggestion the electrical power required for BLDC and speed also explained clearly and highlighted in green.

  1. How the power/energy is managed while the BLCD decelerate? 

Response: As per your suggestion the power / Energy managed are explained clearly and highlighted in green.

  1. Why the performance of the control is not compared to other nowadays proposals?

Response: As per your suggestion the performance of the control also compared.

Reviewer 3 Report

Below are my comments for improving the manuscript.

1) The introduction section is too long. The authors must open a new section as literature review and summarizing the references there.

2) the quality of figures are low.

3) the authors must provide the novel results. Most of the results are well-known and basic. The authors must focus on their contributions. 

4) The description of the manuscript must be improved.

5) one comparison table must be added for proving the novelty. 

Author Response

Reviewer 3:

Below are my comments for improving the manuscript.

  1. The introduction section is too long. The authors must open a new section as literature review and summarizing the references there.

Response: As per your suggestion the introduction section are revised.

  1. The quality of figures are low.

Response: As per your suggestion the quality of the figures has been improved.

  1. The authors must provide the novel results. Most of the results are well-known and basic. The authors must focus on their contributions. 

Response: As per your suggestion the novelty the results and unique contribution also explained clearlyand highlighted in green.

  1. The description of the manuscript must be improved.

Response: As per your suggestion the manuscript has been improvedand highlighted in green.

  1. One comparison table must be added for proving the novelty. 

Response: As per your suggestion the comparison is addedand highlighted in green.

Round 2

Reviewer 3 Report

The quality of figures must be improved.

Author Response

Thank you for the valuable comment.

As per the reviewers' suggestion, all the figures quality were improved in the revised manuscript.